# packetLSTM: Dynamic LSTM Framework for Streaming Data with Varying Feature Space

## Abstract

We study the online learning problem characterized by the varying input feature space of streaming data. Although LSTMs have been employed to effectively capture the temporal nature of streaming data, they cannot handle the dimension-varying streams in an online learning setting. Therefore, we propose a dynamic LSTM-based novel method, called packetLSTM, to model the dimension-varying streams. The packetLSTM's dynamic framework consists of an evolving packet of LSTMs, each dedicated to processing one input feature. Each LSTM retains the local information of its corresponding feature, while a shared common memory consolidates global information. This configuration facilitates continuous learning and mitigates the issue of forgetting, even when certain features are absent for extended time periods. The idea of utilizing one LSTM per feature coupled with a dimension-invariant operator for information aggregation enhances the dynamic nature of packetLSTM. This dynamic nature is evidenced by the model's ability to activate, deactivate, and add new LSTMs as required, thus seamlessly accommodating varying input dimensions. The packetLSTM achieves state-of-the-art results on five datasets, and its underlying principle is extended to other RNN types, like GRU and vanilla RNN.

## 1 Introduction

Online learning, characterized by streaming data, where data instances arrive one by one, has been studied extensively (Gama, 2012; Neu & Olkhovskaya, 2021; Agarwal et al., 2008). Recently, there has been a growing focus on online learning in environments with varying input feature spaces. Examples include movie sentiment classification and crowdedness severity prediction (He et al., 2023; Agarwal et al., 2024). These varying input features, termed haphazard inputs (Agarwal et al., 2023), are denoted as $X^t \in \mathbf{R}^{d^t}$, where $d^t$ indicates the dimensionality of input, varying over time $t$. The field of haphazard inputs is expanding, prompting the introduction of new methods, applications, and appropriate datasets as elaborated in section A of the Appendix.

The current landscape is focused on developing new methods. Predominantly, haphazard inputs are modeled using classical approaches like naive Bayes (Katakis et al., 2005), decision stumps (Schreckenberger et al., 2022; 2023), and linear classifiers (Beyazit et al., 2019), favored for their dynamic architectures. However, there is a push towards developing dynamic deep learning solutions (Agarwal et al., 2022; 2023), motivated by the capabilities of neural networks. Nevertheless, current methodologies have not adequately leveraged the streaming nature of data. To bridge this gap, we advocate using Recurrent Neural Networks (RNNs) (Hochreiter & Schmidhuber, 1997; Zhang et al., 2021a), which can effectively exploit the temporal dynamics of data.

We introduce a novel architecture, termed packetLSTM, designed to dynamically adapt to varying input feature space. This framework employs a unique ensemble of Long Short-Term Memory (LSTM) units, each dedicated to a specific input feature. The packetLSTM allows for robust inter-action among its LSTMs, fostering the integration of global information while preserving feature-specific knowledge within each unit's short-term memory. The packetLSTM facilitates continuous learning without the risk of catastrophic forgetting in online learning environments (Hoi et al., 2021). The feature-based learning, coupled with a dimension-invariant aggregation operator, allows packetLSTM to dynamically activate, deactivate, and add new LSTMs as needed, leading to adeptly managing haphazard inputs. The main contributions of our work are as follows. (1) We intro-

duce the first RNN-based framework, packetLSTM, to effectively handle haphazard inputs in online learning. (2) The packetLSTM exhibits learning without forgetting capabilities in an online learning setting. We demonstrate this capability in a challenging scenario where certain features are absent for extended time periods. (3) The principles underlying the packetLSTM framework are adaptable and can be extended to other types of RNNs, like Gated Recurrent Units (GRUs) and vanilla RNNs. We substantiate this adaptability by developing packetGRU and packetRNN models. (4) We achieve state-of-the-art results across five datasets, as shown in Figure 1. (5) We introduce a strong baseline based on the Transformer called HapTransformer and demonstrate that HapTransformer outperforms other baselines; however, it is still inferior to packetLSTM.

## 2 RELATED WORKS

**Haphazard Inputs** The initial approach to address haphazard inputs utilized naive Bayes and $\chi^2$ statistics to dynamically incorporate features, update existing feature statistics, and select a feature subset (Katakis et al., 2005). This method was further expanded by employing an ensemble of naive Bayes classifiers for predictive analysis (Wenerstrom & Giraud-Carrier, 2006). Subsequent research has focused on inferring unobserved features from observed ones using various techniques, including graph methods (He et al., 2019; Sajedi & Razzazi, 2024), and Gaussian copula (He et al., 2021; Zhuo et al., 2024), followed by the application of classifiers across the complete feature space. Concurrently, another line of research projects data into a shared subspace to learn a linear classifier using empirical risk minimization (Beyazit

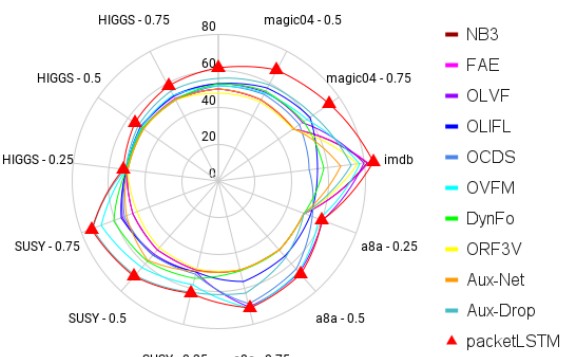

Figure 1: Performance comparison of packetLSTM with other models. The legend and outer labels are methods and datasets, respectively. Exact balanced accuracy values are provided in Table 1.

et al., 2019) or online gradient descent (Zhou & Matsushima, 2023). Distinctly, You et al. (2024) maintains an informativeness matrix of each feature to update a linear classifier. Another research direction explores the use of decision trees to handle haphazard inputs. Specifically, Schreckenberger et al. (2022) proposed Dynamic Forest, which uses an ensemble of decision stumps, each based on a feature from a selected subset of all seen features. However, this approach can result in numerous decision stumps, prompting Schreckenberger et al. (2023) to introduce a refined approach that constructs only one decision stump for each feature. Lee et al. (2023a) proposed to utilize adaptive random forest on a fixed set of features, created through imputation assuming large buffer storage. Despite the dynamic nature of the above classical methods in modifying their architectures, the era of big data necessitates adopting deep learning approaches to effectively model haphazard inputs. To date, seminal contributions in this domain include works by Agarwal et al. (2022; 2023), which are based on neural networks. The Auxiliary Network (Agarwal et al., 2022) incorporates parallel hidden layers to process each input feature. Conversely, Aux-Drop (Agarwal et al., 2023) implements selective and random dropouts within its layers to handle haphazard inputs. Although these models operate under specific assumptions, recent advancements by the same authors (Agarwal et al., 2024) provide simple solutions to mitigate these assumptions in haphazard inputs. While all the methods discussed above handle haphazard inputs, none effectively exploits the temporal dynamics of streaming data, which we achieve using RNNs, specifically LSTMs. The comparison of haphazard inputs with other varying feature space fields is presented in section V of Appendix.

**RNNs** RNNs are among the most popular models in sequential data analysis (Salehinejad et al., 2017; Allen-Zhu & Li, 2019). Various techniques are developed to capture the temporal dynamics of data, including stacking LSTMs to establish a hierarchical framework (Hermans & Schrauwen, 2013; Wang et al., 2017), as well as designing specialized time-modeling LSTM units (Che et al., 2018; Kazemi et al., 2019). In this article, we utilize Time-LSTM (Zhu et al., 2017) as our LSTM units because of its demonstrated capability in modeling both short-term and long-term interest.

LSTMs have also been extensively used in the field of multi-modality (Xie & Wen, 2019; Liu et al., 2018; Xu et al., 2020; Lee et al., 2023b). However, to the best of our knowledge, the RNN-based models proposed in the multi-modal domain or any other domain are not capable of modeling data with varying input dimensions in an online learning setting as discussed in section B of the Appendix. Therefore, we propose a new dynamic RNN framework in this article, filling a significant gap in the research landscape of RNNs and haphazard inputs.

## 3 PRELIMINARIES

**Notations**   In this article, we represent time in superscript and feature id in subscript. For example, $x_2^3$ represents the value of feature 2 ($F_2$) at time $t_3$. For ease of readability, we slightly adjust the notation of time here. Instead of denoting by $v_j^{t_1}$, we use $v_j^1$. When time is referenced individually, it is denoted as $t_1$. Moreover, $t$ indicates a random time, and $t-1$ denotes the time preceding $t$. A complete list of notations is provided in section C of the Appendix.

**Characteristics**   Haphazard inputs exhibit six characteristics which are illustrated in Figure 2(b). These characteristics are: (1) *Streaming data*, which are received sequentially and processed without storage. (2) *Missing data*, which are features present in some instances but may be absent in subsequent ones like $F_1$ at time $t_2$. (3) *Missing features*, that are not received from the onset; however, their availability is known like $F_4$. (4) *Sudden features*, that arrives unexpectedly without prior indication of their existence like $F_3$ at time $t_2$. (5) *Obsolete features*, which can cease to exist after any instance, such as $F_2$. (6) *Unknown number of total features*, results from the combined effect of missing data, missing features, sudden features, and obsolete features.

**Feature Space**   The characteristics of haphazard inputs result in a varying feature space. We define a universal feature space ($\bar{\mathbb{F}}^t$) as the set of features encountered till time $t$. The specific set of features present at time $t$ is represented by $\mathbb{F}^t$ and is termed current feature space as shown in Figure 2(b). The universal feature space will grow with time due to the emergence of sudden features and missing features. For example, $\mathbb{F}^1 = \{F_1, F_2\}$ at time $t_1$, and $\bar{\mathbb{F}^2} = \{F_1, F_2, F_3\}$ at time $t_2$ in Figure 2(b). In an ideal condition, $\bar{\mathbb{F}}^t$ can contract with the removal of obsolete features; however, since the cessation of obsolete features is unknown, $\bar{\mathbb{F}}^t$ may not decrease in practice.

**Mathematical Formulation**   The haphazard input received at time $t$ can be represented by $X^t$, where $X^t \in \mathbf{R}^{|\mathbb{F}^t|}$. Here, $|\cdot|$ represents the cardinality of a set. The corresponding ground truth is denoted by $y^t$, where $y^t \in [0,1]^C$ and $C$ is the number of classes. This paper deals with the binary classification problem but can be easily extended to multi-class scenarios. The model, denoted by $f$, operates in an online learning setting with $f^{t-1} : X^t \to y^t$, where $f^0$ represents the initialized state of the model. After processing $\{X^t, y^t\}$, the model's state is represented by $f^t$. At each time $t$, the model receives $X^t$, and $f^{t-1}$ processes $X^t$ to yield a prediction $\hat{y}^t$. Upon the revelation of $y^t$, the loss $l^t = H(y^t, \hat{y}^t)$ is computed, where $H$ is a loss function. The model then updates from $f^{t-1}$ to $f^t$ for the subsequent instances based on $l^t$. This iterative process continues for each instance.

## 4 METHOD

The packetLSTM consists of a pack of LSTMs, each dedicated to a distinct feature, as illustrated in the gray box in Figure 2(a). We utilize LSTMs due to their proven effectiveness in capturing temporal dynamics of data (Kazemi et al., 2019; Wang et al., 2017; Zhang et al., 2021a).

Due to the varying dimensions of input feature space, a single LSTM cannot process all features, necessitating one LSTM per feature. Each LSTM ($L_j$) receives inputs comprising the feature value ($x_j^t$), time delay ($\Delta_j^t$), its previous short-term memory ($h_j^{t_-}$), and common long-term memory ($c^{t-1}$), as labeled *Input* in Figure 2(a). The $\Delta_j^t$ measures the time difference between the current time $t$ and the last observed time $t_-$ for feature $j$. This temporal information is critical to the model because feature availability varies (Zhu et al., 2017; Che et al., 2018).

At each instance, only LSTMs corresponding to available features are activated. For example, at time $t_2$, the absence of feature $F_1$ results in the deactivation of LSTM $L_1$, depicted by a sketched gray

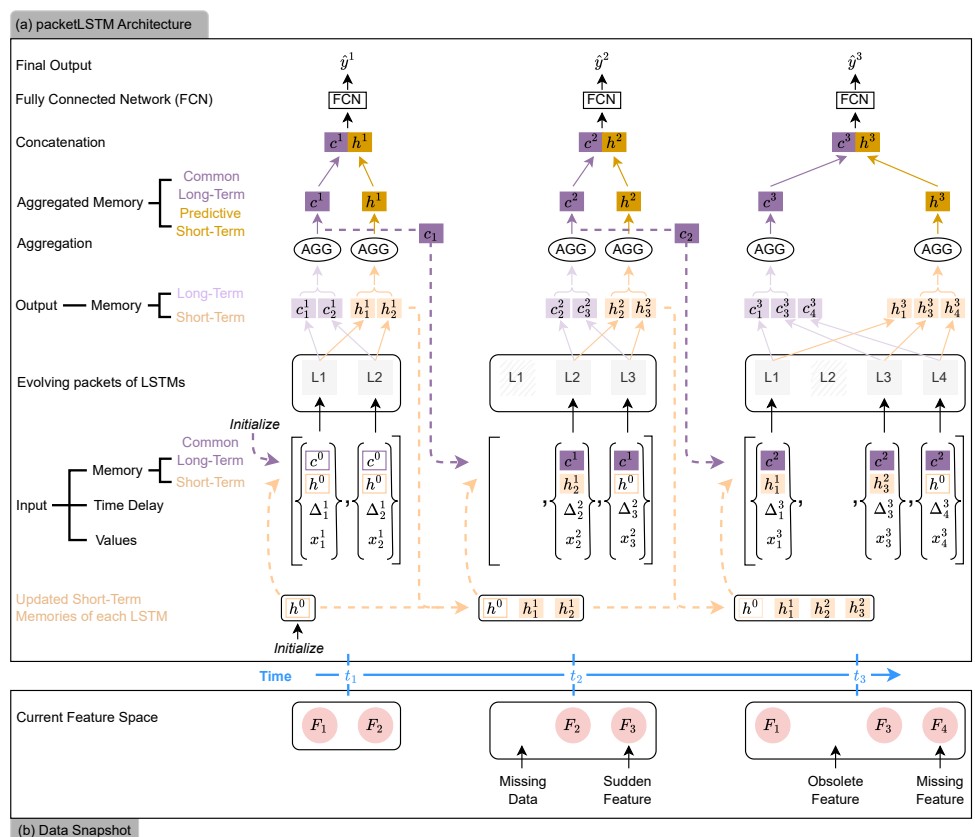

Figure 2: (a) The packetLSTM architecture based on (b) the data snapshot.

box in Figure 2(a). Thus, packetLSTM dynamically handles all the characteristics of the haphazard inputs by activating, deactivating, or adding new LSTMs as needed.

The common long-term memory ($c^t$), which aggregates information from all long-term memories of active LSTMs, facilitates the interaction among features, crucial for enriched knowledge acquisition (Zhang et al., 2021b). This $c^t$ contains global information, while the short-term memory of each LSTM retains the local information about individual features, alleviating the issue of catastrophic forgetting in an online learning setting. The aggregation operator, being dimension-invariant, accommodates the variable number of features. All active short-term memories are aggregated to generate a predictive short-term memory ($h^t$). These memories are subsequently concatenated and processed through a fully connected classifier to produce the final output $\hat{y}^t$ (see Figure 2(a)).

The model parameters are updated based on the loss $l^t = H(y^t, \hat{y}^t)$ in an online learning setting. New LSTMs are introduced with initialized memories ($h^0$, $c^{t-1}$) and a time delay ($\Delta$) set to zero for sudden (e.g., $F_3$ at $t_2$) and missing features (e.g., $F_4$ at $t_3$), as shown in Figure 2.

**Working Principle**    Based on the current feature space $\mathbb{F}^t$ at time $t$, the output of each LSTM is given by $h_j^t, c_j^t = L_j(x_j^t, \Delta_j^t, h_j^{t-}, c^{t-1}) \ \forall \ F_j \in \mathbb{F}^t$. Numerous LSTM variants exist in the literature that model time delay ($\Delta_j^t$), such as decay in GRU-D (Che et al., 2018), time gate in Phased LSTM (Neil et al., 2016), and Time2Vec (Kazemi et al., 2019). In this article, we employ Time-LSTM (Zhu et al., 2017) because of its highly demonstrated capability in modeling time information. Time-LSTM utilizes time delay to capture the short-term interest for current instances and preserves these delays to address long-term interest for future instances. We specifically utilize Time-LSTM 3 (T-3) among its three versions – Time-LSTM 1, Time-LSTM 2, and T-3 – because it integrates the input

and forget gates (Greff et al., 2016), offering a model that is both less computationally intensive and concise. The formulation of T-3 (and other time modeling variants) for feature $j$ at time $t$, within the packetLSTM framework, represented by $L_j$, is discussed in the section D of the Appendix.

Our packetLSTM framework differs from T-3 in its approach to handling input feature spaces. Unlike T-3, which utilizes a single LSTM unit for a fixed feature space, our approach employs a distinct LSTM unit for each feature to manage a varying feature space. This necessitates a different input configuration for each LSTM compared to the T-3. Specifically, rather than using the previous long-term memory of each LSTM as an input, the packetLSTM framework utilizes a common long-term memory, which facilitates interaction among features. Moreover, while T-3 accepts the short-term memory from time $t-1$ as input at time $t$, the packetLSTM inputs the short-term memory from time $t_-$ at time $t$, where $t_-$ is not necessarily equal to $t-1$ in haphazard inputs.

Next, a dimension-invariant aggregation operator ($AGG$) determines the common long-term memory and the predictive short-term memory as

$$c^t = AGG(\bigcup_{j,\forall F_j \in \mathbb{F}^t} \{c_j^t\}), \qquad h^t = AGG(\bigcup_{j,\forall F_j \in \mathbb{F}^t} \{h_j^t\}). \qquad (1)$$

The $AGG$ operator, defined as $AGG : [\mathbf{R}]^{|F^t|} \to \mathbf{R}$, accepts a variable number of inputs, specifically, $|F^t|$ inputs. Given that $c_j^t$ and $h_j^t$ are vectors, the $AGG$ operator performs aggregation element-wise. Common examples of dimension-invariant aggregation operators include mean, maximum, minimum, and summation. Finally, the prediction $\hat{y}^t$ is generated by a fully connected neural network (FCN), applied on the concatenation of $c^t$ and $h^t$ as $\hat{y}^t = \text{FCN}(concat(c^t, h^t))$.

## 5 EXPERIMENTS

**Datasets** We consider 5 datasets – magic04 (Bock et al., 2004), imdb (Maas et al., 2011), a8a (Kohavi et al., 1996), SUSY (Baldi et al., 2014), and HIGGS (Baldi et al., 2014) – with details provided in section E of the Appendix. The motivation to choose these datasets is three-fold. First, they include both real (imdb) and synthetic datasets. Second, the number of instances varies from 19020 in magic04 to 1M in HIGGS. Third, the number of features ranges from 8 in SUSY to 7500 in imdb. Therefore, the diversity in the number of features and instances allows us to determine the efficacy of packetLSTM effectively. The imdb dataset is haphazard in nature. However, the synthetic dataset needs to be transformed into haphazard inputs. Following the baseline papers (Beyazit et al., 2019; Agarwal et al., 2023), we transform synthetic datasets based on the probability values $p$, where $p = 0.25$ means only 25% of features are available at each time instance. We consider $p = 0.25$, 0.5, and 0.75. The synthetic dataset preparation is further discussed in section E of the Appendix.

**Metrics** We compare all models using five metrics: number of errors, accuracy, Area Under the Receiver Operating Characteristic curve (AUROC), Area Under the Precision-Recall Curve (AUPRC), and balanced accuracy. Each metric is discussed in section F of the Appendix. The balanced accuracy is the primary comparison metric in the main manuscript, while detailed comparisons using the other metrics are presented in section K of the Appendix. We adhere to the standard evaluation protocol for haphazard inputs, which is detailed in section G of the Appendix.

**Baseline** We consider 10 baseline models, including NB3(Katakis et al., 2005), FAE (Wenerstrom & Giraud-Carrier, 2006), DynFo (Schreckenberger et al., 2022), ORF$^3$V (Schreckenberger et al., 2023), OLVF (Beyazit et al., 2019), OCDS (He et al., 2019), OVFM (He et al., 2021), Aux-Net (Agarwal et al., 2022), Aux-Drop (Agarwal et al., 2023), and OLIFL(You et al., 2024). Additionally, there are a few other models – see section H of the Appendix – applicable to the field of haphazard inputs. However, we could not include these models because of the lack of open-source code and the challenges associated with implementing them.

**Implementation Details** We ran all the models five times, except for the deterministic models – NB3, FAE, OLVF, and OLIFL – which consistently produce identical outcomes across runs. The results are reported as mean $\pm$ standard deviation (std). For packetLSTM, max and mean aggregation operators were used in synthetic and real datasets, respectively. Additionally, Z-score normalization was implemented in an online manner, as discussed in section 6. The implementation details and hyperparameter search are discussed in the section I and J of the Appendix, respectively.

Table 1: Comparison of models on all datasets based on balanced accuracy. The deterministic models — NB3, FAE, OLVF, and OLIFL — underwent a single execution, and the non-deterministic models were executed 5 times, with the mean $\pm$ standard deviation reported. A [‡] symbol indicates non-deterministic models that were run only once on specific datasets due to substantial time constraints, and [†] denotes the real datasets. % Im. denotes the % improvement in the performance of packetLSTM compared to the previous best-performing baseline (denoted by *italics*) in each dataset.

| Dataset | $p$ | NB3 | FAE | OLVF | OLIFL | OCDS | OVFM | DynFo | ORF$^3$V | Aux-Net | Aux-Drop | **packetLSTM** | % Im. |
|---|---|---|---|---|---|---|---|---|---|---|---|---|---|
| magic04 | 0.25 | 50.01 | 50.01 | 53.18 | 53.06 | 51.89±0.10 | 51.94±0.00 | 52.75±0.30 | 47.94±0.22 | 50.09±0.07 | *56.04±0.53* | **61.33±0.07** | 9.44 |
| | 0.50 | 50.02 | 50.00 | 54.60 | 57.28 | 53.40±0.45 | 54.13±0.08 | 55.12±0.06 | 48.56±0.11 | 50.09±0.03 | *59.29±0.48* | **68.31±0.15** | 15.21 |
| | 0.75 | 49.99 | 50.00 | 56.19 | 60.75 | 53.76±1.07 | 58.79±0.04 | 56.75±0.02 | 49.32±0.04 | 50.05±0.07 | *63.18±0.61* | **73.64±0.11** | 16.56 |
| imdb$^\dagger$ | | 81.56 | *82.18* | 80.08 | 54.36 | 50.14±0.02 | 77.43$^\ddagger$ | 57.98±0.29 | 76.47±0.11 | 67.41$^\ddagger$ | 73.10±0.19 | **85.06±0.04** | 3.50 |
| a8a | 0.25 | 50.01 | 50.00 | **60.67** | 53.57 | 54.75±0.87 | 58.66±0.00 | 50.01±0.03 | 49.99±0.00 | 50.00±0.00 | 50.00±0.01 | 60.53±0.16 | -0.23 |
| | 0.50 | 50.01 | 50.00 | *66.46* | 54.63 | 64.04±1.01 | 66.02±0.00 | 50.11±0.01 | 50.01±0.00 | 50.00±0.00 | 55.33±1.99 | **67.73±0.21** | 1.91 |
| | 0.75 | 50.01 | 50.00 | 70.60 | 56.56 | 68.81±1.10 | *70.95±0.00* | 50.13±0.01 | 49.99±0.00 | 50.00±0.00 | 62.87±0.93 | **71.11±0.16** | 0.23 |
| SUSY | 0.25 | 50.00 | 49.90 | 51.12 | 51.23 | 52.11±0.19 | 58.00±0.00 | 54.69±0.01 | 49.37±0.01 | 50.53±1.17 | *61.98±0.10* | **62.77±0.01** | 1.27 |
| | 0.50 | 50.00 | 50.01 | 53.21 | 53.59 | 54.03±0.28 | 62.85±0.00 | 58.27±0.00 | 48.33±0.02 | 57.89±7.19 | *68.79±0.14* | **69.28±0.01** | 0.71 |
| | 0.75 | 50.00 | 50.12 | 55.98 | 56.39 | 54.84±0.48 | 68.51±0.00 | 60.94±0.01 | 47.53±0.03 | 53.67±8.13 | *73.55±0.11* | **73.85±0.02** | 0.41 |
| HIGGS | 0.25 | 50.00 | 50.16 | 50.57 | 50.56 | 49.97±0.07 | 50.61±0.01 | 50.18$^\ddagger$ | 49.86±0.03 | 49.99±0.00 | *51.17±0.05* | **52.22±0.04** | 2.05 |
| | 0.50 | 50.00 | 50.01 | 51.21 | 51.50 | 50.06±0.06 | 51.40±0.01 | 50.21$^\ddagger$ | 49.82±0.02 | 49.99±0.01 | *53.09±0.05* | **55.47±0.04** | 4.48 |
| | 0.75 | 50.00 | 50.55 | 51.98 | 52.48 | 49.97±0.05 | 52.66±0.00 | 50.16$^\ddagger$ | 49.75±0.03 | 49.98$^\ddagger$ | *55.55±0.11* | **58.71±0.05** | 5.69 |

**Results**   The balanced accuracy ($b$) of all the models is presented in Table 1. The packetLSTM has a statistically significant better performance than all the models across each dataset except in a8a with $p = 0.25$ and 0.75 (hypothesis supported by paired t-tests at 99% significance level). To quantify the performance enhancement of packetLSTM over the previously best-performing baseline ($pbb$), we calculate the % improvement as $(b_{packetLSTM} - b_{pbb}) * 100 / b_{pbb}$, and these results are shown in the last column of Table 1. Among synthetic datasets, packetLSTM surpasses the $pbb$ by at least 9.44% and 2.05% in magic04 and HIGGS, respectively. Moreover, packetLSTM consistently exhibits superior performance across all $p$ within the SUSY dataset. In a8a, packetLSTM achieves the best result in 2 out of 3 cases, except at $p = 0.25$, where its performance is comparable. In the real dataset (imdb), packetLSTM outperforms the $pbb$ by a margin of 3.5%.

## 6 ABLATION STUDIES

We conduct ablation studies within packetLSTM to assess the impact of each component and identify optimal variants, adhering to the hyperparameters described in *Implementation Details* of section 5, unless specified otherwise. We report the mean of the balanced accuracy in the main manuscript (Table 2), with std detailed in section L of the Appendix. We calculate the number of wins for each component across all dataset combinations, defining a win as achieving the highest balanced accuracy or being within a 0.05 margin of the top value, accounting for variability indicated by the std. For example, in the magic04 dataset at $p = 0.25$, the packetLSTM with the Min aggregator reports a slightly higher balanced accuracy (61.36) than the Max (61.33) but exhibits greater std (0.14 vs. 0.07). This variability suggests that some runs using the Min aggregator achieved lower balanced accuracy than the Max. Therefore, a 0.05 margin mitigates such discrepancies.

**Which aggregation operator performs the best?**   The Max operator, in general, performed best with 7 wins, as shown in the AGG component of Table 2. However, to determine the best operators, we compare them one by one (see $\rightarrow$ in Table 2). (1) The performance difference between Min and Max is negligible, indicating both are suitable choices. (2) Between Sum and Mean, there is no evident superiority. However, the Sum is sensitive to the number of features, while the Mean remains relatively stable unless the feature distribution changes substantially. We tested this by alternating $p$ (from 0.25 to 0.75) every 100 instances in the a8a dataset, which has the most features among the synthetic datasets. The Mean operator (balanced accuracy = 62.77) substantially outperformed the Sum operator (54.16), establishing the Mean as the superior operator. (3) Table 2 indicates that the efficacy of the Mean increases as data availability decreases from $p = 0.75$ to 0.25. Further experiments at $p = 0.05$ and 0.95, shown in section M of the Appendix, validate this trend. Consequently, Max is recommended for scenarios with high data availability, while the Mean is preferable otherwise. This is further supported by the imdb dataset ($p = 0.0165$), where the Mean (85.06) significantly surpassed the Max (77.90). Consequently, we utilized the Max aggregator for all the synthetic datasets and the Mean operator for the real dataset.

Table 2: Ablation study on the components (Comp.) of packetLSTM. Here, $w$ stands for the number of wins a variant registered out of all the 13 dataset combinations. The **_variants_** marked in bold and italics are employed in packetLSTM. $\rightarrow$ denotes the difference between two variants.

| Comp. | Variants | magic04 | | | imdb | a8a | | | SUSY | | | HIGGS | | | $w$ |
|---|---|---|---|---|---|---|---|---|---|---|---|---|---|---|---|
| | | 25 | 50 | 75 | | 25 | 50 | 75 | 25 | 50 | 75 | 25 | 50 | 75 | |
| AGG | Mean | 60.98 | 67.06 | 72.30 | **85.06** | 58.52 | 65.15 | 67.50 | **63.16** | **69.59** | 74.07 | 52.01 | 55.27 | 58.04 | 3 |
| | Sum | 61.27 | 67.93 | 73.21 | 83.63 | 58.26 | 65.15 | 68.80 | **63.19** | 69.29 | **74.39** | **52.24** | **55.64** | **59.06** | 5 |
| | Min | **61.36** | **68.28** | 73.58 | 77.64 | **60.53** | **67.78** | **71.13** | 62.78 | 69.29 | 73.85 | **52.21** | 55.50 | 58.70 | 6 |
| | ***Max*** | **61.33** | **68.31** | **73.64** | 77.90 | **60.53** | 67.73 | 71.11 | 62.77 | 69.28 | 73.85 | **52.22** | 55.47 | 58.71 | 7 |
| $\rightarrow$ | Min - Max | 0.03 | -0.03 | -0.06 | -0.26 | 0.00 | 0.05 | 0.02 | 0.01 | 0.01 | 0.00 | -0.01 | 0.03 | -0.01 | |
| $\rightarrow$ | Sum - Mean | 0.29 | 0.87 | 0.91 | -1.43 | 0.30 | 0.00 | 1.30 | 0.03 | -0.30 | 0.32 | 0.03 | 0.37 | 1.02 | |
| $\rightarrow$ | Mean-Max | -0.35 | -1.25 | -1.34 | 7.16 | -2.01 | -2.58 | -3.61 | 0.39 | 0.31 | 0.22 | -0.21 | -0.2 | -0.67 | |
| Time Model | None | 58.83 | 65.12 | 69.87 | 85.18 | 58.77 | 66.52 | 70.30 | 62.30 | 68.70 | 73.25 | 51.37 | 54.05 | 57.06 | 0 |
| | Decay | 58.73 | 64.99 | 69.81 | **85.30** | 58.82 | 67.14 | 70.76 | 62.37 | 68.89 | 73.40 | 51.41 | 54.09 | 57.01 | 1 |
| | TimeLSTM-1 | 61.13 | 68.14 | **73.63** | 85.06 | 60.45 | 67.63 | 71.09 | 62.77 | 69.33 | **73.91** | 52.23 | 55.53 | 58.70 | 8 |
| | TimeLSTM-2 | 61.32 | **68.34** | **73.65** | 85.15 | 60.33 | 67.62 | 71.09 | 62.79 | 69.31 | 73.88 | 52.26 | 55.53 | 58.65 | 9 |
| | ***TimeLSTM-3*** | **61.33** | 68.31 | 73.64 | 85.06 | **60.53** | 67.73 | 71.11 | 62.77 | 69.28 | 73.85 | 52.22 | 55.47 | **58.71** | 10 |
| Feat | Universal | **61.37** | **68.33** | **73.69** | 85.06 | 60.32 | 67.58 | 71.15 | 62.79 | 69.27 | 73.84 | 52.26 | 55.52 | 58.69 | 11 |
| | ***Current*** | 61.33 | 68.31 | 73.64 | 85.06 | **60.53** | **67.73** | **71.11** | 62.77 | 69.28 | 73.85 | 52.22 | 55.47 | 58.71 | 13 |
| Concat | Only LTM | 60.94 | 67.66 | 73.09 | **85.15** | 59.36 | 66.86 | 70.34 | 62.78 | 69.32 | 73.85 | 52.20 | 52.62 | 57.69 | 5 |
| | Only STM | 61.15 | **68.26** | 73.38 | **85.17** | 60.02 | 67.57 | 70.77 | **62.82** | **69.33** | **73.89** | 52.18 | 52.19 | 57.63 | 5 |
| | ***Both*** | **61.33** | **68.31** | **73.64** | 85.06 | **60.53** | **67.73** | **71.11** | 62.77 | 69.28 | 73.85 | **52.22** | **55.47** | **58.71** | 12 |
| Normalize | None | 59.14 | 65.44 | 70.53 | **85.61** | 59.97 | **67.89** | **71.22** | 63.12 | **69.65** | 74.19 | 51.46 | 54.33 | 57.39 | 4 |
| | Min Max | 57.69 | 63.33 | 68.42 | 85.23 | **59.94** | 67.76 | 71.15 | 62.37 | 69.06 | 73.61 | 51.21 | 53.10 | 55.52 | 0 |
| | Decimal Scaling | 55.03 | 60.36 | 65.72 | 85.23 | 49.99 | 50.00 | 50.00 | 50.96 | 58.53 | 63.31 | 50.00 | 50.00 | 50.00 | 0 |
| | Mean Norm | 60.23 | 66.71 | 71.86 | 84.89 | 58.03 | 65.94 | 70.07 | **63.19** | **69.68** | **74.31** | 51.86 | 54.88 | 58.11 | 3 |
| | Unit Vector | 50.01 | 54.23 | 60.02 | 85.32 | 58.12 | 65.67 | 68.97 | 56.52 | 65.43 | 71.44 | 50.75 | 52.86 | 55.53 | 0 |
| | ***Z-score*** | **61.33** | **68.31** | **73.64** | 85.06 | **60.53** | 67.73 | 71.11 | 62.77 | 69.28 | 73.85 | **52.22** | **55.47** | **58.71** | 7 |
| RNN | Vanilla RNN | **60.70** | **66.76** | **72.14** | 83.55 | 51.16 | 65.24 | 64.50 | **62.84** | 68.06 | **73.98** | **51.96** | **54.26** | 56.77 | 7 |
| | GRU | 60.34 | 66.24 | 71.23 | 81.49 | 50.96 | 60.86 | **73.38** | 62.73 | **69.16** | 73.72 | 51.80 | **54.25** | 56.79 | 3 |
| | ***LSTM*** | 58.83 | 65.12 | 69.87 | **85.18** | **58.77** | **66.52** | 70.30 | 62.30 | 68.70 | 73.25 | 51.37 | 54.05 | **57.06** | 4 |

**Is time modeling beneficial?** We investigated 4 variants of time modeling: T-3, Time-LSTM 2, Time-LSTM 1, and Decay (Che et al., 2018). As observed in the `Time Model` component, T-3 performed the best. Moreover, LSTM, without any time modeling (*None*), performed the worst, affirming the beneficial impact of time modeling for haphazard inputs. Moreover, all the 4 variants outperform the $pbb$ in 11 out of 13 scenarios except in a8a ($p = 0.25$) and SUSY ($p = 0.75$).

**Are current feature space sufficient for the predictive capability of packetLSTM?** The packetLSTM utilizes current feature space for the predictive short-term memory. However, it is also possible to consider the universal feature space by aggregating the short-term memories from all LSTMs linked to the universal features for the predictive short-term memory. The current feature space has 13 wins compared to 11 wins of universal feature space (see the `Feat` component). Therefore, the current feature space is sufficient for enhancing the predictive capability of packetLSTM.

**What is the importance of common long-term and predictive short-term memory?** The final prediction in packetLSTM concatenates common long-term and predictive short-term memory. We assessed the importance of each memory type by comparing the original packetLSTM, termed *Both*, with its variants that either exclude predictive short-term memory (*Only LTM*) or common long-term memory (*Only STM*), as shown in the `Concat` component. *Both* significantly outperformed others, securing 12 wins, confirming that both memory types are crucial. While common long-term memory holds global information, predictive short-term memory provides local information, and individually, they surpass $pbb$ in most datasets except in a8a and HIGSS ($p = 0.5$).

**Does the data need to be normalized?** We observed significant variation in the range of feature values across datasets. For example, in magic04, the first feature ranges from 4.28 to 334.18. Therefore, we performed streaming normalization by using five methods: Min-Max, Decimal Scaling, Mean-Norm, Unit Vector, and Z-score, as detailed in section N of the Appendix. Table 2 shows that Z-score performed the best, underscoring the need for streaming normalization in handling haphazard inputs. Remarkably, the packetLSTM, even without any normalization (labeled as *None*), surpassed $pbb$ in all datasets except for a8a ($p = 0.25$), showcasing the superiority of packetLSTM.

**Is the concept of packet architecture adaptable?** The packetLSTM framework's principles are adaptable, leading us to develop packetRNN and packetGRU architectures. We outline these architectures' formulations and performance in section O of the Appendix and the `RNN` component

of Table 2, respectively. Both packetRNN and packetGRU deliver competitive performances compared to $pbb$. Specifically, packetRNN outperforms $pbb$ in the magic04, imdb, SUSY, and HIGGS datasets, while packetGRU excels in magic04, a8a ($p = 0.75$), SUSY, and HIGGS. These results confirm the effectiveness of the packet architecture. We evaluate packetLSTM without any time modeling component for a fair comparison among packet architectures. The packetRNN emerges as the best performer, although it significantly underperforms in the a8a dataset. Consequently, we use LSTM for the packet architecture. Furthermore, LSTM's ability to incorporate long-term memory allows it to encode global information, a characteristic absent in both GRU and vanilla RNN.

**Complexity Analysis** The time complexity of packetLSTM is $\sum_{t=1}^{T} O(g(|\mathbb{F}^t|) * s^2 + P)$, where $T$ is the number of instances, $O(g(|\mathbb{F}^t|) * s^2)$ denotes the complexity for processing all activated LSTMs at time $t$, and $O(P)$ encompasses constant-time operations such as aggregations and final predictions. Here, $s$ is the hidden size of the LSTM, and the function $g(|\mathbb{F}^t|)$ varies between 1 and $|\mathbb{F}^t|$, often close to 1. Thus, the packetLSTM scales well in terms of time complexity with feature size. The space complexity at time $t$ is $O(|\bar{\mathbb{F}}^t| * L + K)$, where $O(L)$ is the space complexity of one LSTM and $O(K)$ accounts for the final prediction network and aggregated memories. The introduction of sudden and missing features increases the space complexity, showing potential scalability limits. However, packetLSTM easily handles the 7500 features in the imdb dataset, affirming its capability to handle high-dimensional data. Detailed calculations of time and space complexities and a strategy to mitigate space complexity are provided in section P and Q of the Appendix.

**packetLSTM vs Single LSTM** The effectiveness of packetLSTM is further evidenced through its comparison with a single LSTM model, which necessitates the availability of all features. This requirement can be met through imputation, although it requires contradicting the sixth characteristic of haphazard inputs where the total number of features is unknown. Nonetheless, for the purpose of evaluating packetLSTM, we compare it against a single LSTM, employing three imputation techniques: forward fill, mean of the last five observed values of a feature, and Gaussian copula (Zhao et al., 2022). Table 3 demonstrates that packetLSTM significantly outperforms single LSTM with all imputation-based techniques across all datasets, except in a8a at $p = 0.75$. Further details on the single LSTM model, hyperparameter search, the Gaussian copula model, and its specific performance issues with the a8a and imdb datasets are discussed in section R of the Appendix.

## 7 CHALLENGING SCENARIOS

We design three challenging experiments to explicitly elucidate the efficacy of packetLSTM to (1) handle sudden features, (2) handle obsolete features, and (3) demonstrate the learning without forgetting capability. We considered the HIGGS and SUSY datasets for these experiments due to their substantial number of instances. In the main manuscript, we discuss the results corresponding to HIGGS, while similar conclusions for SUSY are discussed in section S.2 of the Appendix. For comparative analysis, we opted for OLVF, OLIFL, OVFM, and Aux-Drop baselines based on their superior performance in the HIGGS dataset, as indicated in Table 1. We divided the dataset into 5 intervals, each consisting of 20% of the total instances, with successive intervals containing the next 20% of instances. For the exact values of Figure 3 and 4, refer to section S.1 of the Appendix.

**Sudden Features** Here, each subsequent interval includes an additional 20% of features, starting with 20% in the first interval and increasing to 100% by the fifth interval. This progression, termed the trapezoidal data stream (Zhang et al., 2016; Liu et al., 2022), is illustrated in Figure 3 by a pink-shaded region. Model performance is expected to enhance with increased data volume. All models, except OLVF, exhibit this increasing performance trend. Notably, packetLSTM outperforms other models in each interval, demonstrating its superior capability in handling sudden features.

**Obsolete features** Here, all features are initially present in the first interval, then only the first 80% remain in the second interval, decreasing sequentially as shown in Figure 3. Intuitively, the model's performance should deteriorate as data volume decreases, a pattern observed in all models except Aux-Drop. The Aux-Drop's performance drops significantly in the second interval, suggesting a misleading impression of improvement in the third. The packetLSTM outperforms all other methods in each interval, demonstrating its superior ability to handle obsolete features.

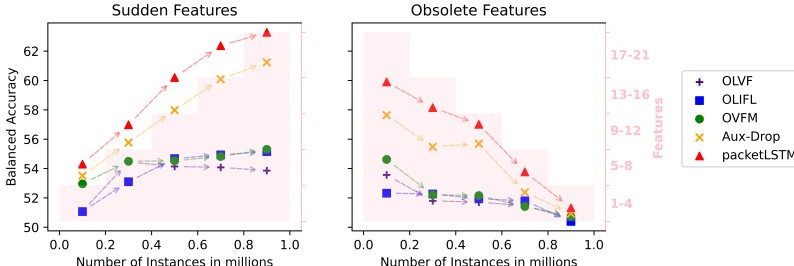

Figure 3: Model performance in the scenario of sudden and obsolete features. The mean balanced accuracy in each data interval (e.g., 0 - 0.2 M) is shown in its middle (0.1 M). The pink-colored y-axis represents the Features. For e.g., '1-4' means features 1 to 4 are available in instances shaded with pink color. Both y-axes are shared by the two graphs.

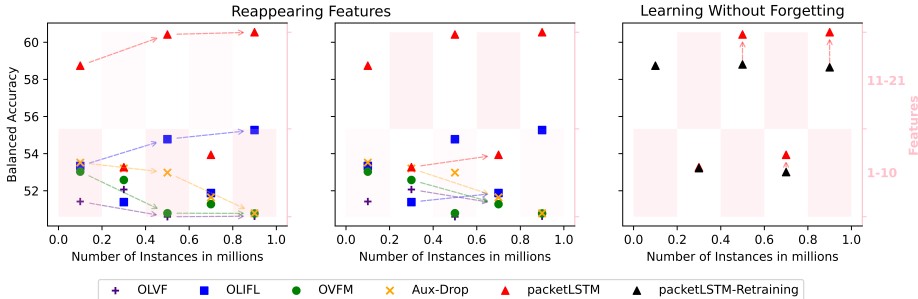

Figure 4: Mitigating catastrophic forgetting: Models performance in reappearing features scenario and its capability of learning without forgetting. The mean balanced accuracy is depicted in the middle of each data interval. Please refer to section S.1 of the Appendix for more details.

**Reappearing**   We assess the model's performance in a scenario where feature sets are disjoint across consecutive intervals and reappear after a gap, as shown in Figure 4. The first 50% of features are present in the first, third, and fifth intervals, while the remaining 50% appear in the second and fourth intervals. Ideally, effective learning without forgetting would result in improved performance upon the reappearance of the same features. However, apart from packetLSTM and OLIFL, all models show declining performance as they progress from the first to the third and fifth intervals and from the second to the fourth interval. Even OLIFL fails in the SUSY dataset as its performance declines from the second to the fourth interval (see section S.2 of the Appendix). The packetLSTM's ability to retain knowledge is attributed to each feature's local information stored in its corresponding LSTM's short-term memory. To quantify knowledge retention, we compare packetLSTM with a version retrained (packetLSTM-retraining) at each interval using the available features (see the rightmost graph of Figure 4). The packetLSTM's balanced accuracy improves by 1.61 and 1.89 at the third and fifth intervals, respectively, and by 0.93 at the fourth interval, demonstrating effective learning without forgetting. Notably, there is also a performance increase in the second interval despite those features not being previously observed. This increase is due to the global information learned during the first interval, aiding subsequent predictions. Overall, packetLSTM outperforms all other baselines in each interval. Further discussion on mitigating catastrophic forgetting is provided in section S.1 of the Appendix. It is important to distinguish that catastrophic forgetting within an online learning setting differs from online continual learning (see section T of the Appendix).

## 8   TRANSFORMER ON HAPHAZARD INPUTS

Despite the lack of application of the Transformer (Vaswani et al., 2017) in the field of haphazard inputs, its inherent ability to manage variable-size inputs makes it a natural choice. Therefore, we also investigate Transformer-based methodologies for modeling haphazard inputs.

Table 3: Comparison of packetLSTM vs single LSTM with imputation, transformer with padding, Set Transformer, and HapTransformer on all datasets based on balanced accuracy.

| Dataset | $p$ | Single LSTM + Imputation | | | Transformer + Padding | | Set Transformer | HapTransformer | **packetLSTM** |
|---------|-----|-------|----------|----------|-------------|-------|-----------------|----------------|----------------|
| | | Ffill | KNN Mean | Gaussian | *Only Values* | *Pairs* | | | |
| magic04 | 0.25 | 52.56±0.33 | 57.42±0.07 | 57.82±0.10 | 56.41±0.14 | 53.95±0.07 | 57.27±0.11 | 60.25±0.25 | **61.33±0.07** |
| | 0.50 | 59.86±0.07 | 64.01±0.12 | 64.52±0.11 | 58.66±0.19 | 56.61±0.12 | 59.50±0.07 | 66.62±0.14 | **68.31±0.15** |
| | 0.75 | 68.97±0.10 | 70.91±0.10 | 71.32±0.05 | 63.93±0.08 | 60.09±0.14 | 62.04±0.79 | 72.11±0.25 | **73.64±0.11** |
| imdb | | 50.42±0.06 | 50.37±0.07 | - | 55.58±0.17 | 51.09±0.26 | 51.06±0.12 | 69.40±0.28 | **85.06±0.04** |
| a8a | 0.25 | 50.12±0.08 | 54.12±0.18 | - | 49.98±0.01 | 50.00±0.00 | 50.00±0.00 | 57.76±0.23 | **60.53±0.16** |
| | 0.50 | 59.54±0.15 | 65.16±0.26 | - | 50.02±0.02 | 50.00±0.00 | 50.00±0.00 | 65.95±0.21 | **67.73±0.21** |
| | 0.75 | 68.74±0.20 | **71.31±0.31** | - | 50.05±0.02 | 50.00±0.00 | 50.00±0.00 | 69.37±0.10 | 71.11±0.16 |
| SUSY | 0.25 | 57.20±0.01 | 59.31±0.03 | 59.59±0.01 | 58.03±0.03 | 58.78±0.11 | 58.17±0.03 | 62.31±0.02 | **62.77±0.01** |
| | 0.50 | 64.16±0.02 | 66.26±0.03 | 66.87±0.02 | 62.22±0.10 | 63.97±0.06 | 62.03±0.01 | 69.13±0.01 | **69.28±0.01** |
| | 0.75 | 71.01±0.02 | 72.18±0.01 | 72.85±0.02 | 67.60±0.13 | 69.59±0.07 | 65.16±0.03 | 73.74±0.03 | **73.85±0.02** |
| HIGGS | 0.25 | 50.54±0.01 | 51.42±0.02 | 51.77±0.22 | 50.43±0.01 | 50.29±0.01 | 50.12±0.01 | 51.82±0.05 | **52.22±0.04** |
| | 0.50 | 52.92±0.02 | 54.54±0.03 | 54.84±0.02 | 51.07±0.02 | 50.58±0.01 | 50.28±0.01 | 55.18±0.04 | **55.47±0.04** |
| | 0.75 | 56.76±0.02 | 58.27±0.03 | 58.39±0.02 | 52.58±0.03 | 51.13±0.02 | 50.38±0.01 | 58.44±0.04 | **58.71±0.05** |

**Padding** We consider padding inputs with zeros or truncating them, which necessitates specifying a fixed input length ($f_l$). If the number of features in an instance ($|\mathbb{F}^t|$) is less than $f_l$, the Transformer pads the input with zeros, and if $|\mathbb{F}^t|$ exceeds $f_l$, it truncates the excess features, potentially leading to information loss. A potential, albeit inefficient, solution is to set an excessively high $f_l$. Nonetheless, to assess how packetLSTM compares to Transformer, we conducted two experiments: *Only Values*, padding available feature values, and *Pairs*, pairing each feature value with its feature ID and padding the sequence. The packetLSTM outperforms Transformer with padding across all dataset scenarios (see Table 3). Further details can be found in section U.1 of the Appendix.

**Natural Language** Given the variable size of inputs, natural language can be seen as an application where features arrive one by one, and most are missing. We compared packetLSTM with DistilBERT (Sanh et al., 2019) and BERT (Devlin et al., 2019), detailed in section U.2 of the Appendix. We observe that both DistilBERT and BERT are unable to perform classification on haphazard inputs with a balanced accuracy of around 50 in all cases.

**Set Transformer** The Set Transformer (Lee et al., 2019), designed for variable-length inputs, has been previously used in offline learning. We employ it for online learning to handle haphazard inputs. Results in Table 3 indicate that packetLSTM significantly outperforms Set Transformer across all datasets, possibly due to Set Transformer's assumption of permutation invariance, which does not hold for haphazard inputs. More details are provided in section U.3 of the Appendix.

**HapTransformer** To tackle permutation invariance, we introduce HapTransformer, which transforms each feature into a distinct learnable embedding, a technique similarly utilized in prior research (Huang et al., 2020; Gorishniy et al., 2021; Somepalli et al., 2021). However, these models do not accommodate variable-sized inputs. The learnable embeddings are subsequently processed by the Set Transformer's decoder, allowing implicit communication of feature identities. Details on the architecture and hyperparameters are provided in section U.4 of the Appendix. Despite its strengths, packetLSTM surpasses HapTransformer in all dataset scenarios (see Table 3). HapTransformer struggles particularly with datasets having high feature counts like imdb and a8a, and requires significant computational time. However, it remains a strong baseline, outperforming other baseline models in the SUSY and HIGGS datasets, as detailed in Tables 1 and 3.

## 9 CONCLUSION

In conclusion, our work introduces packetLSTM, a dynamic framework for handling streaming data with varying feature dimensions in real-time learning environments. Significantly, the underlying principles of packetLSTM are extendable to other architectures, such as vanilla RNN and GRUs, demonstrating the model's adaptability to different neural architectures. The packetLSTM not only outperforms existing methods across various datasets but also showcases flexibility in adapting to changing data conditions without forgetting previously learned information. Our research opens new avenues for further exploration of dynamic neural architectures and sets a new benchmark for online learning with haphazard inputs.

## 10 ETHICS STATEMENT

The packetLSTM framework introduces a novel method for handling streaming data with variable features, potentially impacting numerous fields such as healthcare, finance, autonomous systems, environmental monitoring, and personalized recommendation systems. In healthcare, it can improve real-time patient monitoring by adapting to new or absent data types, enhancing patient care. Financial sectors can benefit from more stable predictive models for trading and risk management due to their ability to process erratic market data. Autonomous systems can gain reliability by effectively managing inconsistencies in sensory data, while environmental monitoring may achieve greater accuracy in tracking ecological changes, aiding policy decisions. In digital platforms, it may refine personalized recommendations by adjusting to user behavior changes, and in educational technology, it may personalize content to student needs, improving engagement and learning outcomes.

Due to the potential application of packetLSTM in the above-discussed crucial and sensitive fields, it is necessary to consider ethical concerns. The packetLSTM framework must navigate several ethical issues to align with the European Union's Artificial Intelligence Act (EU AI Act). Deploying packetLSTM technology requires careful consideration, including data privacy, transparency in decision-making, and addressing data biases to prevent discrimination and ensure fairness across diverse user groups. By addressing these challenges, packetLSTM can significantly enhance efficiency and personalization across multiple domains while upholding high ethical standards.

## 11 REPRODUCIBILITY

All the information to reproduce the result is available in sections 4, 5, 6, 7, and 8 of the main manuscript. Additional information is also provided in sections I, J, R, and U of the Appendix. The code is attached to this submission in the supplementary material with sufficient instructions to faithfully reproduce all the experimental results. The link to the datasets is provided in section E of the Appendix. We report the resources used to conduct the experiments in section I of the Appendix. Moreover, for the benchmark results, we also report the time taken by each individual model in section K of the Appendix.

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

## A  PRACTICAL APPLICATIONS

With the progress of the field of haphazard inputs, more datasets are expected to become available. For instance, Schreckenberger et al. (2023) introduced the crowdsense dataset in 2023, collected from environmental sensors (measuring sound pressure, eCO2 level, and eTVOC level) across Spain's 56 largest cities over 790 days from January 1st (2020) to February 28th (2022). The feature space varies as the new sensing data continuously emerges while many old sensors stop to provide data. The aim is to predict government restriction severity based on sensed regional crowdedness. We could not include this dataset in our study because it has only 790 instances, which is very small for a deep-learning model. Nevertheless, we anticipate the emergence of larger datasets in this domain. Note that we demonstrate the superior performance of packetLSTM compared to article (Schreckenberger et al., 2023) in our study in Table 1 in the main manuscript.

Another potential application is in the study of sub-cellular organisms, as discussed in (Agarwal et al., 2024). The mitochondria undergo morphological changes during the drug discovery process, leading to fusion, fission, and kiss-and-run events. Quantifying these events is crucial for biologists, yet direct observation is impractical due to hundreds of mitochondria within a single cell. Fusion and fission introduce obsolete and sudden features, while kiss-and-run events can lead to sudden, obsolete, and missing features. The combination of haphazard input and segmentation models may effectively model the mitochondrial dynamics.

## B  MULTI-MODAL LEARNING VS PACKETLSTM

The multi-modality requires handling data from different data domains, like images, audio, etc. Some studies in the multi-model domain utilize LSTMs (more generally RNNs), as discussed below.

**Brain Image Segmentation**  LSTM-MA (Xie & Wen, 2019) converts the multi-modal image slices to feature sequences using pixel and superpixel constraints. These are then fed to the LSTM, followed by a fully connected neural network to predict the final class label. Finally, the segmentation image is obtained by combining all the classified nodes of a slice.

**Online Action Detection**  Liu et al. (2018) proposed an RNN framework for online action detection and forecasting on the fly from the untrimmed stream data. The data have two modalities, i.e., RGB video and skeleton sequence data. The paper uses deep ConvNet and motion networks to extract embedding from RGB and skeleton data, respectively, and the embeddings are processed individually by stacked LSTM.

**Speech Recognition**  AE-MSR (Xu et al., 2020) works on two modalities, namely, audio and video, to perform speech recognition. Here, both audio and video data are passed to an AE subnetwork to generate visual features and enhanced audio magnitude. The generated features are further passed to different element-wise attention-gated recurrent units (EleAtt-GRU) encoders. This generates visual and audio context, respectively. Both contexts are further passed to a decoder consisting of EleAtt-GRU to generate outputs.

Within the multi-modal domain, to the best of our knowledge, the total number of features/modalities always remains the same and is known at the outset ($t = 0$). Some recent work also considers missing modality in the multi-modal domain (Lee et al., 2023b), but still, the total number of modalities is known at the outset ($t = 0$). Therefore, the above method cannot accommodate haphazard inputs. Whereas the packetLSTM can dynamically change its architecture to adapt to varying feature spaces in real-time environments.

## C  NOTATIONS

All the notations used in this article are provided in Table 4 with their corresponding meanings.

Table 4: Notations and their meaning in this article.

| Notations | Meaning |
|---|---|
| $t$ | A random time or the $t^{\text{th}}$ instance |
| $X^t$ | The set of input values arrived at time $t$ |
| $y^t$ | Ground truth at time $t$ |
| $C$ | Number of output classes |
| $\mathbf{R}$ | Real numbers |
| $d^t$ | The number of input features arrived at time $t$ |
| $t$-1 | The instance preceding time $t$ |
| $F_j$ | Feature $j$ |
| $\bar{\mathbb{F}}^t$ | Universal feature space |
| $\mathbb{F}^t$ | Current feature space |
| $f^t$ | The trained model after time $t$ |
| $\hat{y}^t$ | Prediction at time $t$ by model $f$ |
| $l^t$ | Loss of the model $f$ at time $t$ |
| $H$ | Loss function |
| $L_j$ | LSTM corresponding to feature $j$ |
| $t_k$ | The $k^{\text{th}}$ time or the instance |
| $x_j^k$ | The value of feature $j$ at time $t_k$ |
| $\Delta_j^t$ | Time delay of feature $j$ at time $t$ |
| $h_j^t$ | Short-term memory of $L_j$ at time $t$ |
| $c_j^t$ | Long-term memory of $L_j$ at time $t$ |
| $h_j^{t_-}$ | Last observed short-term memory of $L_j$ before time $t$ |
| $h^t$ | Predictive short-term memory at time $t$ |
| $c^t$ | Common long-term memory at time $t$ |
| $h^0$ | Initialized values of short-term memory |
| $c^0$ | Initialized values of long-term memory |
| $i_j^t$ | Input gate at time $t$ of $L_j$ |
| $T1_j^t$ | Time gate 1 at time $t$ of $L_j$ |
| $T2_j^t$ | Time gate 2 at time $t$ of $L_j$ |
| $o_j^t$ | Output gate at time $t$ of $L_j$ |
| $\tilde{c}_j^t$ | Cell state at time $t$ of $L_j$ |
| $W, w, b$ | Weight parameters of LSTM |
| $s$ | Size of the short-term and long-term memory |
| $\sigma_{v \in \{c,h\}}$ | Tanh function |
| $\sigma_{v \in \{t,T1,T2,\Delta,o\}}$ | Sigmoid functions |
| $f_l$ | fixed input length |
| $|\mathbb{F}^t|$ | Number of available features at time $t$ |

## D DIFFERENT TIME-MODELING VARIANTS WITHIN THE CONTEXT OF PACKETLSTM

In this article, we present the packetLSTM framework with four different time modeling variants. Primarily, we employ Time-LSTM 3 (Zhu et al., 2017) within the packetLSTM framework. Additionally, we explore the use of Time-LSTM 1, Time-LSTM 2, and decay (Che et al., 2018) into the packetLSTM architecture. The following paragraphs discuss all these time-modeling variants within the context of packetLSTM.

**Time-LSTM 3** The formulation of Time-LSTM 3 (and other time modeling variants) for feature $j$ at time $t$, within the packetLSTM framework, represented by $L_j$, is given by $h_j^t, c_j^t =$

$L_j(x_j^t, \Delta_j^t, h_j^{t-}, c^{t-1}) \; \forall \; F_j \in \mathbb{F}^t$. Mathematically, the computation of $L_j$ is expressed as

$$i_j^t = \sigma_i(x_j^t W_{j,xi} + h_j^{t-} W_{j,hi} + c^{t-1} \odot w_{j,ci} + b_{j,i}), \tag{2}$$

$$T1_j^t = \sigma_{T1}(x_j^t W_{j,xT1} + \sigma_\Delta(\Delta_j^t W_{j,T1}) + b_{j,T1}), \tag{3}$$

$$T2_j^t = \sigma_{T2}(x_j^t W_{j,xT2} + \sigma_\Delta(\Delta_j^t W_{j,T2}) + b_{j,T2}), \tag{4}$$

$$\tilde{c}_j^t = (1 - i_j^t \odot T1_j^t) \odot c^{t-1} + i_j^t \odot T1_j^t \odot \sigma_c(x_j^t W_{j,x\tilde{c}} + h_j^{t-} W_{j,h\tilde{c}} + b_{j,\tilde{c}}), \tag{5}$$

$$c_j^t = (1 - i_j^t) \odot c^{t-1} + i_j^t \odot T2_j^t \odot \sigma_c(x_j^t W_{j,xc} + h_j^{t-} W_{j,hc} + b_{j,c}), \tag{6}$$

$$o_j^t = \sigma_o(x_j^t W_{j,xo} + \Delta_j^t W_{j,o} + h_j^{t-} W_{j,ho} + \tilde{c}_j^t \odot w_{j,\tilde{c}o} + b_{j,o}), \tag{7}$$

$$h_j^t = o_j^t \odot \sigma_h(\tilde{c}_j^t). \tag{8}$$

where $i_j^t, T1_j^t, T2_j^t$, and $o_j^t$ represents the input gate, time gate 1, time gate 2, and output gate of the LSTM $L_j$ at time $t$, respectively. The cell state $\tilde{c}_j^t$ influences the current prediction through the output gate and the short-term memory $h_j^t$. The functions $\sigma_i, \sigma_{T1}, \sigma_{T2}, \sigma_\Delta$, and $\sigma_o$ are sigmoid functions, while $\sigma_c$ and $\sigma_h$ are tanh functions. The $W_{j,hi}$ denotes the weight associated with the short-term memory within the input gate of $L_j$. This definition extends analogously to all $W$ parameters. The symbol $\odot$ indicates the Hadamard product, and all the $w$ parameters are peephole connection weights (Gers & Schmidhuber, 2000).

**Time-LSTM 2**   Similar to Time-LSTM 3, Time-LSTM 2 has two gates. Therefore, equations 2, 3, 4, 7, and 8 hold true for Time-LSTM 2, along with an addition of forget gate ($f_j^t$), an update of the equation of cell state ($\tilde{c}_j^t$), and long-term memory ($c_j^t$).

$$f_j^t = \sigma_f(x_j^t W_{j,xf} + h_j^{t-} W_{j,hf} + c^{t-1} \odot w_{j,cf} + b_{j,f}), \tag{9}$$

$$\tilde{c}_j^t = f_j^t \odot c^{t-1} + i_j^t \odot T1_j^t \odot \sigma_c(x_j^t W_{j,x\tilde{c}} + h_j^{t-} W_{j,h\tilde{c}} + b_{j,\tilde{c}}), \tag{10}$$

$$c_j^t = f_j^t \odot c^{t-1} + i_j^t \odot T2_j^t \odot \sigma_c(x_j^t W_{j,xc} + h_j^{t-} W_{j,hc} + b_{j,c}). \tag{11}$$

**Time-LSTM 1**   Time-LSTM 1 only contains a single time gate. Therefore, the formulation of Time-LSTM 1 within the packetLSTM framework is given by equations 2, 9, 3, 10, 7, and 8.

**Decay**   The decay mechanism introduced by Che et al. (2018) attenuates the short-term memory by a learnable factor $\gamma_j^t$. Subsequently, the process adheres to the conventional operations of a vanilla LSTM, which is given by

$$
\begin{aligned}
\gamma_j^t &= exp\{-max(0, W_{j,\gamma}\Delta_j^t + b_{j,\gamma})\}, \\
\tilde{h}_j^{t-} &= \gamma_j^t \odot h_j^{t-}, \\
i_j^t &= \sigma_i(x_j^t W_{j,xi} + \tilde{h}_j^{t-} W_{j,hi} + b_{j,i}), \\
f_j^t &= \sigma_f(x_j^t W_{j,xf} + \tilde{h}_j^{t-} W_{j,hf} + b_{j,f}), \\
\tilde{c}_j^t &= \sigma_c(x_j^t W_{j,x\tilde{c}} + \tilde{h}_j^{t-} W_{j,h\tilde{c}} + b_{j,\tilde{c}}), \\
c_j^t &= f_j^t c^{t-1} + i_j^t \tilde{c}_j^t, \\
o_j^t &= \sigma_o(x_j^t W_{j,xo} + \tilde{h}_j^{t-} W_{j,ho} + b_{j,o}), \\
h_j^t &= o_j^t \sigma_h(c_j^t).
\end{aligned}
\tag{12}
$$

# E   DATASETS

The detailed descriptions of each dataset are provided in Table 5 and further elaborated below:

- **magic04 (Bock et al., 2004)**: It is a Monte Carlo simulated dataset for the registration of high-energy particles in an atmospheric Cherenkov telescope. The binary classification

Table 5: Dataset Description: # {Instances, Features} represents the number of {Instances, Features}.

| Dataset | # Instances | # Features | Imbalance | Missing Values % | Type |
|---------|-------------|------------|-----------|------------------|------|
| magic04 | 19020 | 10 | 64.84% | 25%, 50%, 75% | Synthetic |
| imdb | 25000 | 7500 | 50% | 98.35% | Real |
| a8a | 32561 | 123 | 75.92% | 25%, 50%, 75% | Synthetic |
| SUSY | 1M | 8 | 45.79% | 25%, 50%, 75% | Synthetic |
| HIGGS | 1M | 21 | 52.97% | 25%, 50%, 75% | Synthetic |

task associated with magic04 is to distinguish between a shower image caused by primary gammas (1) and cosmic rays in the upper atmosphere (0). The magic04 dataset can be accessed via this link[1].

- **imdb (Maas et al., 2011)**: It is a movie sentiment classification dataset with labels as positive (1) or negative (0). The original imdb dataset consists of training and test subsets, each containing 25000 instances. Following the previous literature in the field of haphazard inputs (Beyazit et al., 2019; Agarwal et al., 2024), we consider the training subset of the data provided in this link[2] and the 7500 most prevalent words within this subset, which represents the 7500 features. This approach adheres to the standard practices of previous works, ensuring fair comparison.

- **a8a (Kohavi et al., 1996)**: It is the income data from a census conducted in 1994. The task is to classify where the income exceeds $50k per year. The dataset is pre-processed, resulting in 123 features. The a8a dataset can be accessed via this link[3].

- **SUSY (Baldi et al., 2014)**: It is a Monte Carlo simulated data of kinematic properties of particles with a binary classification task of predicting between a signal process (1) where SUSY particles are produced and a background process (0) with the same detectable particles. Following article (Agarwal et al., 2023), the first 8 features are considered. The SUSY data can be accessed via this link[4].

- **HIGGS (Baldi et al., 2014)**: Similar to SUSY, it is a Monte Carlo simulation data associated with a binary classification task to differentiate between a signal process (1) where new theoretical Higgs bosons are produced and a background process (0) with identical decay products but distinct kinematic features. We consider the first 21 features of the HIGGS dataset (Agarwal et al., 2023). The HIGGS data can be accessed via this link[5].

**Synthetic Dataset Preparation**   We created haphazard input datasets from synthetic datasets based on probability values $p$, as defined in the baseline papers (Beyazit et al., 2019; Agarwal et al., 2023). Specifically, $100 \times p\%$ of features at each time instance is simulated as available independently of each other following a uniform distribution. For example, if $p = 0.25$, 25% of features at each time instance is only available. From each synthetic dataset, three subsets are generated corresponding to $p$ values of 0.25, 0.5, and 0.75. This approach creates a spectrum of datasets, from highly unavailable data to those with extensive data availability, thereby facilitating the testing of models under varying conditions of data accessibility. Notably, 98.35% of data values are unavailable in the imdb dataset (see Table 5).

## F   METRICS

**Number of Errors**   This is defined as the number of instances incorrectly classified by the model. A model is deemed more effective when it exhibits a lower number of errors. However, note that the number of errors may not be a suitable metric for imbalanced data.

---

[1] https://archive.ics.uci.edu/dataset/159/magic+gamma+telescope
[2] https://ai.stanford.edu/~amaas/data/sentiment/
[3] https://www.csie.ntu.edu.tw/~cjlin/libsvmtools/datasets/binary.html
[4] https://archive.ics.uci.edu/dataset/279/susy
[5] https://archive.ics.uci.edu/dataset/280/higgs

**Accuracy**   The Model's accuracy can be determined by calculating the total number of correct predictions divided by the total number of instances. Similar to the number of errors, accuracy may not serve as a reliable metric in the context of imbalanced datasets, as it can be misleadingly high when the majority class is predicted correctly while neglecting the minority class.

**AUROC**   The Area Under the Receiver Operating Characteristic Curve (AUROC) measures a model's capability to differentiate between positive and negative classes. The ROC curve plots the true positive rates against the false positive rates. The area of this curve, the AUROC, is a value for which higher numbers indicate superior model performance.

**AUPRC**   AUPRC stands for Area Under the Precision-Recall Curve. It is similar to AUROC; however, it utilizes precision and recall rather than true and false positive rates. AUPRC is particularly valuable for assessing the performance of models on imbalanced datasets, providing insight into the model's ability to identify positive instances.

**Balanced Accuracy**   This is calculated as the average of specificity and sensitivity. This metric offers a more nuanced assessment of model performance across imbalanced datasets compared to traditional accuracy, as it equally weighs the correct identification rates of both positive and negative classes.

## G   EVALUATION PROTOCOL

We adhere to the standard evaluation protocol for haphazard inputs and, more generally, online learning. As described in the *Mathematical Formulation* paragraph of Section 3 in the main manuscript, the model is evaluated and then trained iteratively. There is no separate evaluation set. The model first receives input features at time $t$, makes a prediction, and then the ground truth is revealed. The model calculates the cross-entropy loss using the prediction and ground truth. Each instance is processed once and not revisited, reflecting the online learning characteristics of batch size 1 and epoch 1. The prediction logits and the labels for each instance are stored, and upon processing all the inputs, the balanced accuracy and other metrics are determined based on these data. This evaluation method is consistently applied across all models in this study.

## H   BASELINES NOT INCLUDED FOR COMPARISON

The following models apply to the field of haphazard inputs: OLCF (Zhou & Matsushima, 2023), OIL (Lee et al., 2023a), DCDF2M (Sajedi & Razzazi, 2024), OFSVF (Zhuo et al., 2024), DFLS (Li & Gu, 2023), RSOL (Chen et al., 2024), OVFIV (Qin & Song, 2024), RAIL (Kim et al., 2024), OLBD (Yan et al., 2024b), OLCDS (Zhou et al., 2024), and OLIDS$_{PLM}$ (Yan et al., 2024a). However, the inclusion of these additional models in our article was not feasible due to the lack of open-source code and the challenges associated with implementing them, which stemmed from either insufficient detail or the complexity of the models.

## I   IMPLEMENTATION DETAILS

All the models were implemented using the PyTorch framework, and the experiments were conducted on an NVIDIA DGX A100 machine. Since the model operates in an online learning setting, CPUs were exclusively used to sequentially process the data.

**packetLSTM**   The size for both the long-term and short-term memory components was set at 64. The final fully connected network is a two-layer neural network with a ReLU non-linear activation function. Stochastic gradient descent was employed for back-propagation. Cross-entropy loss and AdamW optimizer were used. The learning rate for magic04, imdb, a8a, SUSY, and HIGGS is set as 0.0006, 0.0008, 0.0009, 0.0008, and 0.0002, respectively.

**Baselines**   In the case of Aux-Drop baseline (Agarwal et al., 2023), the online deep learning framework was adopted as the foundational model. The implementation strategies for the majority of

Table 6: Description of all the hyperparameters used in each model and their search values. This table is adapted from article (Agarwal et al., 2024). {} represents individual values, and [] represents a range.

| Model | Hyperparameter(s) | Description | Search |
|---|---|---|---|
| NB3 | n | proportion of total features to consider as the number of top features | {.2, .4, .6, .8, 1} |
| FAE | m | number of instances after which learner's output is considered | 5 |
| | f | threshold on the difference of youngest learner's feature set and current top M features | 0.15 |
| | p | number of consecutive instances a learner is under the threshold before being removed | 3 |
| | r | minimum number of instances between the 2 consecutive learners | 10 |
| | N | number of instances to calculate accuracy over | 50 |
| | M | proportion of total features to train new feature forest on | {.2, .4, .6, .8, 1} |
| OLVF | C | loss bounding parameter for instance classifier | {.0001, .01, 1} |
| | $\bar{C}$ | loss bounding parameter for feature classifier | {.0001, .01, 1} |
| | B | proportion of selected features for sparsity | {.01, .1, .3, .5, .7, .9, 1} |
| | $\lambda$ | regularization parameter | {.0001, .01, 1} |
| OCDS | T | number of instances after which 'p' (weighing factor) is updated | {8, 16} |
| | $\alpha$ | absorption scale parameter used in equation 10 of (He et al., 2019) | {.0001, .001, .01, .1, 1} |
| | $\beta_0$ | absorption scale introduced by us for the 1st term in eq. 9 of He et al. (2019) | {.0001, .001, .01, .1, 1} |
| | $\beta_1$ | tradeoff parameter used in equation 9 of (He et al., 2019) | {.0001, .001, .01, .1, 1} |
| | $\beta_2$ | tradeoff parameter used in equation 9 of (He et al., 2019) | {.0001, .001, .01, .1, 1} |
| OVFM | decay_choice (dc) | decay update rules choices (see original code (He et al., 2021)) | {0, 1, 2, 3, 4} |
| | contribute_error_rate (ce) | used in the original code implementation of classifiers in (He et al., 2021) | {.001, .005, .01, .02, .05} |
| | window_size (ws) | a (buffer-like) window to store data instances | 20 |
| | decay_coef_change (dc) | set 'True' for learning rate decay, 'False' otherwise | {True, False} |
| | batch_size_denominator (bs) | used in update step in case of learning rate decay | {8, 10, 20} |
| | batch_c (bc) | added to the denominator for stability in learning rate decay | 8 |
| DynFo | $\alpha$ | impact on weight update | {.1, .5} |
| | $\beta$ | probability to keep the weak learner in ensemble | {.5, .8} |
| | $\delta$ | fraction of features to consider (bagging parameter) | {.001, .01} |
| | $\epsilon$ | penalty if the split of decision stump is not in the current instance | {.001, .01} |
| | $\gamma$ | threshold for error rate | {.5, .8} |
| | M | number of learners in the ensemble | {500, 1000} |
| | N | buffer size of instances | 20 |
| | $\theta1$ | lower bounds for the update strategy | .05 |
| | $\theta2$ | upper bounds for the update strategy | {.6, .75} |
| ORF3V | forestSize (fS) | number of stumps for every feature | {3, 5, 10} |
| | replacementInterval (rI) | instances after which stumps might get replaced | {5, 10} |
| | updateStrategy (uS) | strategy to replace stumps | {oldest, random} |
| | replacementChance (rC) | probability to not replace stump for "random" update strategy | .7 |
| | windowsize (ws) | buffer storage size on which to determine feature statistics | 20 |
| | $\alpha$ | weight update parameter | {.01, .1, .3, .5, .9} |
| | $\delta$ | pruning parameter | .001 |
| Aux-Net | n_base_layer (nb) | number of base layers | 5 |
| | n_end_layers (ne) | number of end layers | 5 |
| | n_nodes_layers (n) | number of nodes in each layer | 50 |
| | lr | learning rate | {.001, .005, .01, .05, .1, .3, .5} |
| | b | discount rate | .99 |
| | s | smoothing parameter | .2 |
| Aux-Drop | max_num_hidden_layers (nl) | number of hidden layers | 6 |
| | neuron_per_hidden_layer (n) | number of nodes in each hidden layer except the AuxLayer | 50 |
| | n_neuron_aux_layer (na) | total number of neurons in the AuxLayer | $\sim 5 \times$num_feat |
| | b | discount rate | .99 |
| | s | smoothing parameter | .2 |
| | lr | learning rate | {.001, .005, .01, .05, .1, .3, .5} |
| | dropout_p (d) | dropout rate in the AuxLayer | {.3, .5} |
| OLIFL | option | The options to determine $\tau$, where loss* = loss/inner_product(X) | {loss*, min(C, 2loss*)} |
| | C | Tradeoff parameter of loss | [1e-6, 10] |
| **packetLSTM** | hidden_size | size of short-term and long-term memory | {32, 64} |
| | lr | learning rate | [0.05, 0.0001] |
| | aggregate_by (agg) | dimension-invariant aggregation function | {Mean, Max, Min, Sum} |

baseline models were in accordance with the guidelines provided by Agarwal et al. (2024), except for the OLIFL model (You et al., 2024), for which the implementation from the OLIFL article was utilized. The OVFM model (He et al., 2021), which typically requires buffer storage to store the inputs and thus contradicts the principles of online learning, was adapted by limiting the buffer storage to two instances.

Table 7: Best hyperparameters of the models based on balanced accuracy. In synthetic datasets, we consider $p = 0.5$ for hyperparameter search. Experiments marked with $\S$ employed heuristically set hyperparameters without a search process due to the high computational demands. The best value of $\lambda$ for OLVF and hidden_size for packetLSTM is always found to be .0001 and 64, respectively, for all the datasets.

| Models | magic04 | imdb | a8a | SUSY | HIGGS |
|---|---|---|---|---|---|
| NB3 (n) | 0.6 | 0.4 | 0.2 | 1.0 | 0.2 |
| FAE (M) | 1.0 | 0.8 | 0.2 | 0.6 | 0.4 |
| OLVF (C, $\bar{C}$, B) | .0001, .0001, 1 | .01, .0001, 1 | 1, .0001, 1 | .01, .01, 1 | .01, .0001, 1 |
| OLIFL (option, C) | 0, - | 1, .007 | 1, 2 | 1, .3 | 1, .08 |
| OCDS (T, $\alpha$, $\beta_0$, $\beta_1$, $\beta_2$) | 16, .01, .01, .0001, .0001 | 16, 1, .0001, .01, .01$^\S$ | 16, 1, 1, .0001, .0001 | 16, .0001, .01, .0001, .0001 | 8, .0001, .0001, .01, .0001 |
| OVFM (dc, ce, dc, bs) | 3, .005, F, 20 | 4, .001, F, 20$^\S$ | 4, .001, F, 20$^\S$ | 4, .001, F, 20$^\S$ | 4, .001, F, 20$^\S$ |
| DynFo ($\alpha$, $\beta$, $\delta$, $\epsilon$, $\gamma$, M, $\theta2$) | .5, .5, .1, .001, .7, 1000, .6 | .5, .8, .001, .001, .7, 1000, .6$^\S$ | .5, .5, .03, .001, .7, 1000, .6 | .5, .5, .4, .001, .7, 1000, .6$^\S$ | .5, .5, .2, .001, .7, 1000, .6$^\S$ |
| ORF$^3$V (fS, rI, uS, $\alpha$) | 10, 5, random, .01 | 10, 5, oldest, .01$^\S$ | 10, 10, oldest, .1 | 5, 10, random, .1$^\S$ | 5, 10, random, .1$^\S$ |
| Aux-Net (lr) | .5 | .01$^\S$ | .01$^\S$ | 0.05$^\S$ | 0.05$^\S$ |
| Aux-Drop (na, lr, d) | 100, .01, .3 | 30000, .01, .3$^\S$ | 500, .01, .3 | 100, .05, .3$^\S$ | 100, .05, .3$^\S$ |
| **packetLSTM** (lr, agg) | .0006, max | .0008, mean | .0009, max | .0008, max | .0002, max |

## J   HYPERPARMETER SEARCHING

We followed the strategy employed in article (Agarwal et al., 2024), where the best hyperparameters for the synthetic dataset are found at $p = 0.5$ and subsequently used for $p = 0.25$ and 0.75. The best hyperparameters for all baseline models, except OLIFL, are derived from article (Agarwal et al., 2024), and the same hyperparameter search protocol is applied to both OLIFL and packetLSTM. Details of the hyperparameter search values and the selected best values are presented in Table 6 and Table 7. The hyperparameter search process for packetLSTM and OLIFL is discussed next.

**packetLSTM**   We determine the best values of hyperparameters sequentially. First, we fixed the learning rate and aggregation operator to 0.0001 and mean, respectively, and varied the hidden sizes of the short-term and long-term memory, being set at 32 and 64. We found the best hidden size to be 64. Next, we experimented with four aggregation operators, namely, mean, max, min, and sum, maintaining a fixed learning rate of 0.0001 and a hidden size of 64. We found max and mean as the best operators for synthetic and real datasets, respectively. Finally, with the best hidden size and aggregation operator established, we tested a range of learning rates (0.05, 0.01, 0.005, 0.001, 0.0005, 0.0001). We searched in the vicinity of the optimal learning rate found in the previous step. For example, 0.0005 was found to yield good results on the magic04 dataset, prompting further exploration of nearby values – 0.0002, 0.0003, 0.0004, 0.0006, 0.007, 0.0008, and 0.0009. Among these, 0.0006 emerged as the most effective learning rate. The same process was followed for all the datasets.

**OLIFL**   We fixed the option second (min(C, 2loss/inner_product(X))) to determine $\tau$ and tested a range of $C$ values (10, 1, 0.1, 0.01, 0.001, 0.0001, 0.00001, 0.000001). Similar to packetLSTM, we searched in the vicinity of the optimal $C$ value found in the previous step. In addition to this, we also experimented with the first option (loss/inner_product(X)) to determine $\tau$.

## K   BENCHMARKING RESULTS ON OTHER METRICS

In the main manuscript, we compared the models in terms of their balanced accuracy. We also provide the comparison of models on other metrics and time in Table 8.

## L   ABLATIONS STUDIES RESULTS WITH STANDARD DEVIATION

The ablation study conducted in section 6 of the main manuscript reports only the mean balanced accuracy across five runs, as shown in Table 2. The comprehensive results, including both the mean and the standard deviation, are presented in Table 9.

Table 8: Comparison of models on all the metrics and their execution time. The deterministic models — NB3, FAE, OLVF, and OLIFL — underwent a single execution. In contrast, the non-deterministic models were executed 5 times, with the mean ± standard deviation reported. The bAcc, Acc, ROC, PRC, Err, and Time stand for balanced accuracy, accuracy, AUROC, AUPRC, number of errors, and execution time, respectively. A ‡ symbol indicates non-deterministic models that were run only once on specific datasets due to substantial time constraints.

| Dataset | p | Metric | NB3 | FAE | OLVF | OLIFL | OCDS | OVFM | DynFo | ORF$^3$V | Aux-Net | Aux-Drop | **packetLSTM** |
|---|---|---|---|---|---|---|---|---|---|---|---|---|---|
| magic04 | 0.25 | bAcc | 50.01 | 50.01 | 53.18 | 53.06 | 51.89±0.10 | 51.94±0.00 | 52.75±0.30 | 47.94±0.22 | 50.09±0.07 | 56.04±0.53 | 61.33±0.07 |
| | | Acc | 64.81 | 64.84 | 59.59 | 57.22 | 62.99±0.03 | 65.04±0.00 | 59.04±0.37 | 37.96±0.19 | 63.31±0.42 | 67.63±0.30 | 70.95±0.06 |
| | | ROC | 49.91 | 48.66 | 44.02 | 46.93 | 48.79±0.26 | 57.22±0.00 | 50.29±0.15 | 48.00±0.09 | 50.13±0.25 | 57.99±0.76 | 67.79±0.08 |
| | | PRC | 64.73 | 62.53 | 59.34 | 68.63 | 62.96±0.22 | 70.14±0.00 | 65.56±0.11 | 63.04±0.05 | 64.80±0.19 | 68.46±0.59 | 77.48±0.09 |
| | | Err | 6694 | 6687 | 7686 | 8137 | 7039.60±5.13 | 6650.00±0.00 | 7789.40±69.64 | 11800.00±36.69 | 6978.40±79.97 | 6156.60±57.79 | 5524.60±10.63 |
| | | Time | 1.46 | 72.7 | 2.09 | 2.16 | 4.30±0.01 | 70.19±0.09 | 46.59±5.63 | 952.28±23.98 | 376.94±3.42 | 111.51±7.59 | 69.58±5.50 |
| | 0.50 | bAcc | 50.02 | 50.00 | 54.6 | 57.28 | 53.40±0.45 | 54.13±0.08 | 55.12±0.06 | 48.56±0.11 | 50.09±0.03 | 59.29±0.48 | 68.31±0.15 |
| | | Acc | 64.81 | 64.83 | 61.06 | 58.83 | 54.26±2.00 | 65.02±0.04 | 64.12±0.06 | 35.06±0.12 | 63.38±0.60 | 68.91±0.37 | 75.25±0.13 |
| | | ROC | 50.51 | 47.97 | 45.38 | 42.71 | 55.43±1.54 | 61.15±0.06 | 53.24±0.04 | 47.27±0.07 | 49.90±0.34 | 62.39±0.34 | 76.65±0.12 |
| | | PRC | 65.21 | 61.36 | 60.29 | 67.54 | 66.40±1.20 | 72.72±0.06 | 67.65±0.02 | 62.24±0.09 | 64.61±0.22 | 71.25±0.31 | 83.01±0.15 |
| | | Err | 6693 | 6690 | 7407 | 7831 | 8699.00±380.40 | 6653.80±8.14 | 6824.40±10.95 | 12351.40±22.33 | 6965.60±113.62 | 5912.80±70.86 | 4707.00±23.94 |
| | | Time | 1.6 | 72.16 | 2.69 | 2.18 | 5.44±0.65 | 76.92±11.83 | 1110.78±38.39 | 1109.44±17.12 | 478.10±30.42 | 116.15±1.19 | 71.77±1.93 |
| | 0.75 | bAcc | 49.99 | 50 | 56.19 | 60.75 | 53.76±1.07 | 58.79±0.04 | 56.75±0.02 | 49.32±0.04 | 50.05±0.07 | 63.18±0.61 | 73.64±0.11 |
| | | Acc | 64.82 | 64.83 | 62.31 | 62.67 | 54.83±2.14 | 61.83±0.02 | 66.58±0.01 | 34.82±0.03 | 63.32±0.72 | 70.66±0.38 | 78.78±0.10 |
| | | ROC | 49.34 | 46.5 | 46.89 | 39.24 | 55.90±1.74 | 60.95±0.02 | 53.79±0.04 | 44.26±0.13 | 50.16±0.24 | 68.48±0.92 | 82.64±0.14 |
| | | PRC | 64.15 | 59.67 | 61.4 | 64.53 | 67.60±1.39 | 77.80±0.05 | 67.60±0.01 | 59.68±0.14 | 64.88±0.33 | 75.93±0.43 | 87.38±0.16 |
| | | Err | 6691 | 6689 | 7168 | 7101 | 8591.00±406.63 | 7259.00±4.36 | 6356.60±1.21 | 12397.40±6.58 | 6977.20±136.68 | 5580.80±72.82 | 4035.2±18.94 |
| | | Time | 1.78 | 51.8 | 2.16 | 2.09 | 3.20±0.00 | 62.21±7.17 | 1134.59±13.96 | 1259.35±21.46 | 693.89±19.41 | 122.99±2.65 | 70.85±2.45 |
| imdb | | bAcc | 81.56 | 82.18 | 80.08 | 54.36 | 50.14±0.02 | 77.43‡ | 57.98±0.29 | 76.47±0.11 | 67.41‡ | 73.10±0.19 | 85.06±0.04 |
| | | Acc | 81.56 | 82.18 | 80.08 | 54.36 | 50.14±0.02 | 77.43‡ | 57.98±0.29 | 76.47±0.11 | 67.41‡ | 73.10±0.19 | 85.06±0.04 |
| | | ROC | 88.08 | 48.17 | 48.57 | 45.64 | 50.04±0.01 | 83.92‡ | 43.11±0.14 | 55.53±0.02 | 71.34‡ | 79.67±0.29 | 92.66±0.01 |
| | | PRC | 84.93 | 47.95 | 48.29 | 59.91 | 50.13±0.02 | 82.50‡ | 45.68±0.15 | 55.28±0.01 | 71.27‡ | 77.74±0.40 | 92.49±0.02 |
| | | Err | 4609 | 4455 | 4980 | 11410 | 12464.20±6.14 | 5643‡ | 10504.00±73.27 | 5882.80±27.38 | 8148‡ | 6725.60±48.12 | 3734.8±11.12 |
| | | Time | 1257.01 | 4117.33 | 22.17 | 1570.06 | 47818.07±2783.21 | 109199.99‡ | 4072.86±138.29 | 2768.24±23.26 | 223699.04‡ | 46437.42±18114.58 | 4734.65±153.79 |
| a8a | 0.25 | bAcc | 50.01 | 50 | 60.67 | 53.57 | 54.75±0.87 | 58.66±0.00 | 50.01±0.03 | 49.99±0.00 | 50.00±0.00 | 50.00±0.01 | 60.53±0.16 |
| | | Acc | 75.9 | 75.91 | 72.19 | 75.78 | 73.07±0.34 | 78.00±0.00 | 75.73±0.02 | 75.90±0.00 | 75.91±0.00 | 75.85±0.07 | 77.93±0.03 |
| | | ROC | 51.98 | 49.7 | 63.25 | 46.27 | 61.04±1.41 | 77.70±0.00 | 50.86±0.09 | 47.88±0.05 | 54.45±0.18 | 55.92±3.39 | 75.78±0.19 |
| | | PRC | 77.2 | 75.82 | 85.07 | 66.67 | 82.31±0.88 | 91.36±0.00 | 76.35±0.06 | 74.49±0.03 | 77.78±0.23 | 81.27±2.08 | 90.13±0.09 |
| | | Err | 7847 | 7843 | 9056 | 7886 | 8768.60±109.75 | 7165.00±0.00 | 7903.00±6.71 | 7846.00±0.00 | 7843.80±0.45 | 7865.00±22.56 | 7184.6±10.91 |
| | | Time | 22.94 | 25.7 | 4.96 | 37.16 | 19.17±0.02 | 2139.21±88.33 | 4394.51±72.96 | 207.23±4.25 | 5484.63±54.75 | 421.91±119.74 | 240.96±4.76 |
| | 0.50 | bAcc | 50.01 | 50.00 | 66.46 | 54.63 | 64.04±1.01 | 66.02±0.00 | 50.11±0.01 | 50.01±0.00 | 50.00±0.00 | 55.33±1.99 | 67.73±0.21 |
| | | Acc | 75.92 | 75.91 | 76.93 | 76.47 | 74.23±0.64 | 80.42±0.00 | 75.84±0.01 | 75.91±0.00 | 75.91±0.00 | 77.34±0.55 | 80.64±0.12 |
| | | ROC | 51 | 49.74 | 69.64 | 45.36 | 73.56±0.97 | 83.74±0.00 | 51.96±0.08 | 47.28±0.04 | 58.66±0.65 | 72.34±2.56 | 82.07±0.18 |
| | | PRC | 76.55 | 75.84 | 88.6 | 60.82 | 89.40±0.51 | 94.12±0.00 | 76.88±0.06 | 74.27±0.02 | 79.40±0.26 | 89.28±0.93 | 92.97±0.09 |
| | | Err | 7842 | 7843 | 7512 | 7661 | 8390.60±207.37 | 6377.00±0.00 | 7865.40±3.65 | 7844.00±0.00 | 7843.80±0.84 | 7377.00±178.05 | 6302.2±37.75 |
| | | Time | 23.52 | 26.18 | 6.59 | 42.82 | 51.67±3.29 | 2989.01±178.88 | 4858.85±133.63 | 395.44±0.80 | 17639.23±2014.52 | 235.54±8.39 | 243.05±4.79 |
| | 0.75 | bAcc | 50.01 | 50 | 70.63 | 56.56 | 68.81±1.10 | 70.95±0.00 | 50.13±0.01 | 49.99±0.00 | 50.00±0.00 | 62.87±0.93 | 71.11±0.16 |
| | | Acc | 75.92 | 75.91 | 79.66 | 77.70 | 74.57±0.73 | 82.48±0.00 | 75.85±0.01 | 75.88±0.00 | 75.91±0.00 | 79.94±0.34 | 82.20±0.12 |
| | | ROC | 50.88 | 49.74 | 72.43 | 43.44 | 77.80±0.57 | 86.79±0.00 | 52.04±0.08 | 46.94±0.05 | 62.44±0.51 | 79.66±1.14 | 84.96±0.09 |
| | | PRC | 76.16 | 75.84 | 90 | 55.82 | 91.64±0.31 | 95.35±0.00 | 77.11±0.06 | 74.07±0.03 | 81.26±0.30 | 92.35±0.43 | 94.20±0.05 |
| | | Err | 7842 | 7843 | 6622 | 7271 | 8279.80±238.28 | 5705.00±0.00 | 7863.00±4.06 | 7852.00±0.00 | 7843.40±0.55 | 6533.00±111.55 | 5796.20±37.90 |
| | | Time | 28.16 | 37.84 | 6.37 | 48.37 | 32.28±3.32 | 3712.20±459.57 | 5263.24±92.75 | 569.20±11.54 | 28776.93±2045.19 | 577.31±768.95 | 246.70±6.71 |
| SUSY | 0.25 | bAcc | 50 | 49.9 | 51.12 | 51.23 | 52.11±0.19 | 58.00±0.00 | 54.68±0.01 | 49.37±0.01 | 50.53±1.17 | 61.98±0.10 | 62.77±0.01 |
| | | Acc | 54.2 | 47.09 | 49.59 | 50.09 | 50.26±0.30 | 60.06±0.00 | 54.85±0.01 | 50.49±0.01 | 54.39±0.93 | 63.79±0.09 | 64.29±0.01 |
| | | ROC | 49.88 | 50.41 | 56.26 | 48.77 | 55.93±0.32 | 57.52±0.01 | 50.18±0.00 | 49.42±0.01 | 51.03±1.80 | 69.26±0.12 | 69.68±0.01 |
| | | PRC | 45.68 | 46.06 | 57.31 | 54.51 | 53.64±0.94 | 59.45±0.00 | 49.54±0.00 | 45.23±0.01 | 46.93±2.26 | 67.50±0.09 | 68.64±0.01 |
| | | Err | 457957 | 529137 | 504062 | 499125 | 497382.00±2960.08 | 399356.00±16.84 | 451496.67±75.94 | 495106.00±134.17 | 456143.60±9290.25 | 362096.60±871.73 | 357084.2±63.48 |
| | | Time | 68.97 | 12100.93 | 105.85 | 97.67 | 163.96±1.52 | 3771.36±499.81 | 654852.68±8956.39 | 24252.71±37.64 | 16455.42±167.84 | 5773.05±129.01 | 3538.95±32.03 |
| | 0.50 | bAcc | 50 | 50.01 | 53.21 | 53.59 | 54.03±0.28 | 62.85±0.00 | 58.27±0.00 | 48.33±0.02 | 57.89±7.19 | 68.79±0.14 | 69.28±0.01 |
| | | Acc | 54.2 | 46.87 | 51.6 | 52.70 | 51.77±0.36 | 64.36±0.00 | 58.76±0.00 | 48.42±0.03 | 60.71±6.08 | 69.90±0.12 | 70.25±0.01 |
| | | ROC | 50.03 | 50.22 | 61.86 | 46.41 | 57.95±0.32 | 64.34±0.00 | 50.04±0.00 | 49.06±0.01 | 60.14±8.30 | 76.48±0.15 | 76.78±0.01 |
| | | PRC | 45.79 | 45.51 | 62.65 | 53.28 | 54.01±0.65 | 66.74±0.00 | 45.88±0.00 | 44.85±0.01 | 57.49±10.17 | 75.44±0.10 | 76.19±0.02 |
| | | Err | 457957 | 531338 | 483988 | 472973 | 482360.60±3558.98 | 356403.2±18.85 | 412369.00±36.87 | 515823.80±285.70 | 392908.20±60770.38 | 300952.40±1155.92 | 297545.2±145.48 |
| | | Time | 75.86 | 10731.91 | 135.96 | 96.68 | 255.58±0.50 | 3418.06±393.33 | 343724.64±5176.04 | 23928.85±169.87 | 19612.72±567.55 | 6054.62±660.84 | 3722.70±81.00 |
| | 0.75 | bAcc | 50 | 50.12 | 55.98 | 56.39 | 54.84±0.48 | 68.51±0.00 | 60.94±0.01 | 47.53±0.03 | 53.67±8.13 | 73.55±0.11 | 73.85±0.02 |
| | | Acc | 54.2 | 47.04 | 54.72 | 56.05 | 52.24±0.54 | 69.71±0.00 | 61.61±0.01 | 47.50±0.03 | 57.06±6.93 | 74.40±0.08 | 74.63±0.02 |
| | | ROC | 49.96 | 49.51 | 65.15 | 43.61 | 58.41±0.32 | 72.28±0.00 | 49.91±0.00 | 48.36±0.02 | 55.41±10.29 | 81.07±0.07 | 81.42±0.02 |
| | | PRC | 45.75 | 46.43 | 65.74 | 53.12 | 53.39±0.40 | 74.04±0.00 | 45.8±0.00 | 44.28±0.02 | 50.87±10.38 | 80.44±0.11 | 81.35±0.02 |
| | | Err | 457952 | 529567 | 452821 | 439520 | 477624.00±5394.23 | 302922.6±10.21 | 383919.4±129 | 524984.20±345.24 | 429402.00±69311.90 | 255994.60±801.85 | 253676.6±186.74 |
| | | Time | 83.59 | 12324.28 | 111.37 | 96.24 | 166.07±0.20 | 2578.27±161.13 | 114179.9±2433.89 | 25690.16±79.67 | 23987.22±1326.00 | 5787.78±123.95 | 4188.41±86.62 |
| HIGGS | 0.25 | bAcc | 50 | 50.16 | 50.57 | 52.48 | 49.97±0.05 | 50.61±0.01 | 50.18‡ | 49.86±0.03 | 49.99±0.00 | 51.17±0.05 | 52.22±0.04 |
| | | Acc | 52.96 | 51.98 | 51.74 | 52.32 | 49.96±0.05 | 52.58±0.01 | 50.73‡ | 49.68±0.03 | 52.72±0.02 | 53.57±0.02 | 54.15±0.03 |
| | | ROC | 50.03 | 50.09 | 49.99 | 49.44 | 49.96±0.07 | 51.13±0.00 | 49.99‡ | 50.00±0.01 | 50.04±0.02 | 52.36±0.17 | 54.27±0.03 |
| | | PRC | 52.97 | 52.98 | 53.1 | 56.91 | 52.82±0.06 | 53.80±0.00 | 52.98‡ | 53.02±0.01 | 53.01±0.02 | 54.93±0.16 | 56.58±0.03 |
| | | Err | 470403 | 480197 | 482630 | 476766 | 500440.40±475.98 | 474238.00±73.15 | 492724‡ | 503204.00±312.46 | 472809.80±233.06 | 464302.00±186.18 | 458531.4±279.81 |
| | | Time | 136.12 | 17777.05 | 112.34 | 199.83 | 201.20±12.85 | 8122.34±230.07 | 1845662.94‡ | 42814.64±94.35 | 28363.61±153.32 | 5784.12±159.85 | 4396.83±69.00 |
| | 0.50 | bAcc | 50 | 50.01 | 51.21 | 53.06 | 50.06±0.06 | 51.40±0.01 | 50.21‡ | 49.82±0.02 | 49.99±0.01 | 53.09±0.05 | 55.47±0.04 |
| | | Acc | 52.96 | 52.56 | 52.22 | 53.06 | 50.06±0.05 | 53.09±0.01 | 50.78‡ | 48.77±0.02 | 52.73±0.04 | 54.81±0.04 | 56.46±0.02 |
| | | ROC | 49.97 | 50.12 | 50.29 | 48.50 | 50.04±0.08 | 52.42±0.00 | 50.01‡ | 49.98±0.01 | 50.05±0.01 | 55.60±0.06 | 58.59±0.02 |
| | | PRC | 52.9 | 53 | 53.29 | 56.59 | 52.86±0.05 | 54.86±0.00 | 52.99‡ | 52.96±0.01 | 53.01±0.02 | 57.81±0.12 | 60.62±0.03 |
| | | Err | 470388 | 474407 | 477760 | 477973 | 499377.20±487.22 | 469145.00±96.03 | 492219‡ | 512329.40±189.51 | 472704.40±358.46 | 435406.2±242.82 | |
| | | Time | 142.97 | 7885.78 | 145.83 | 193.74 | 267.35±0.54 | 8241.69±231.79 | 1788308.41‡ | 42623.97±97.07 | 44123.61±283.06 | 6039.45±565.24 | 4500.17±165.67 |
| | 0.75 | bAcc | 50 | 50.55 | 51.98 | 52.48 | 49.97±0.05 | 52.66±0.00 | 50.16‡ | 49.75±0.03 | 49.98‡ | 55.55±0.11 | 58.71±0.05 |
| | | Acc | 52.97 | 49.24 | 52.75 | 53.77 | 49.97±0.05 | 53.73±0.00 | 50.78‡ | 47.84±0.03 | 52.72‡ | 56.77±0.08 | 59.25±0.05 |
| | | ROC | 49.95 | 50.1 | 50.66 | 47.52 | 49.98±0.02 | 54.01±0.00 | 49.94‡ | 49.88±0.01 | 50.06‡ | 58.96±0.18 | 62.78±0.05 |
| | | PRC | 52.94 | 53.48 | 53.66 | 56.83 | 52.80±0.02 | 56.07±0.00 | 52.91‡ | 52.89±0.00 | 53.04‡ | 60.78±0.22 | 64.53±0.04 |
| | | Err | 470336 | 507591 | 472523 | 462305 | 500300.00±475.25 | 462742.00±14.27 | 492196‡ | 521169.00±301.39 | 472790‡ | 432298.00±791.48 | 407498.6±538.70 |
| | | Time | 153.05 | 549606.27 | 125.45 | 191.47 | 200.58±2.95 | 8095.05±218.48 | 801655.55‡ | 44079.06±134.05 | 65500.41‡ | 5762.40±170.11 | 5186.53±50.48 |

# M    MEAN VS MAX AGGREGATION OPERATOR

Table 10 presents the comparison between the Mean and Max aggregation operators in the packetLSTM framework. It is evident from Table 10 that the performance of the Mean operator relative to the Max increases as data availability decreases from $p = 0.95$ to $0.05$. The best hyperparameters found at $p = 0.5$ is used for $p = 0.05$ and $0.95$.

# N    STREAMING NORMALIZATION

The details of each streaming normalization technique are discussed below.

Table 9: Ablation study on the components (Comp.) of packetLSTM architecture. This is the copy of Table 2 with standard deviation (std). The results are reported as mean ± std of 5 runs. This result corresponds to the HIGGS dataset.

| Comp. | Variants | magic04 | | | imdb | a8a | | | SUSY | | | HIGGS | | |
|---|---|---|---|---|---|---|---|---|---|---|---|---|---|---|
| | | 25 | 50 | 75 | | 25 | 50 | 75 | 25 | 50 | 75 | 25 | 50 | 75 |
| Avg | Mean | 60.98±0.08 | 67.06±0.10 | 72.30±0.07 | 85.06±0.04 | 58.52±0.21 | 65.15±0.14 | 67.50±0.19 | 63.16±0.02 | 69.59±0.01 | 74.07±0.05 | 52.01±0.01 | 55.27±0.04 | 58.04±0.02 |
| | Sum | 61.27±0.07 | 67.93±0.14 | 73.21±0.20 | 83.63±0.13 | 58.26±0.56 | 65.15±0.14 | 68.80±1.21 | 63.19±0.01 | 69.29±0.01 | 74.39±0.00 | 52.24±0.02 | 55.64±0.01 | 59.06±0.02 |
| | Min | 61.36±0.14 | 68.28±0.08 | 73.58±0.10 | 77.64±0.21 | 60.53±0.27 | 67.78±0.31 | 71.13±0.12 | 62.78±0.01 | 69.29±0.01 | 73.85±0.00 | 52.21±0.02 | 55.50±0.03 | 58.70±0.07 |
| | Max | 61.33±0.07 | 68.31±0.15 | 73.64±0.10 | 77.90±0.23 | 60.53±0.16 | 67.73±0.20 | 71.11±0.15 | 62.77±0.01 | 69.28±0.01 | 73.85±0.02 | 52.22±0.04 | 55.47±0.04 | 58.71±0.05 |
| Time Models | None | 58.83±0.12 | 65.12±0.14 | 69.87±0.12 | 85.18±0.08 | 58.77±0.33 | 66.52±0.22 | 70.30±0.25 | 62.30±0.01 | 68.70±0.00 | 73.25±0.02 | 51.37±0.02 | 54.05±0.07 | 57.06±0.09 |
| | Decay | 58.73±0.20 | 64.99±0.27 | 69.81±0.09 | 85.30±0.06 | 58.82±0.49 | 67.14±0.14 | 70.76±0.20 | 62.37±0.03 | 68.89±0.06 | 73.40±0.18 | 51.41±0.03 | 54.09±0.06 | 57.01±0.06 |
| | TimeLSTM-1 | 61.13±0.09 | 68.14±0.07 | 73.63±0.14 | 85.06±0.06 | 60.45±0.14 | 67.63±0.13 | 71.09±0.17 | 62.77±0.01 | 69.33±0.01 | 73.91±0.01 | 52.23±0.04 | 55.53±0.06 | 58.70±0.06 |
| | TimeLSTM-2 | 61.32±0.07 | 68.34±0.19 | 73.65±0.19 | 85.15±0.06 | 60.33±0.28 | 67.62±0.22 | 71.09±0.26 | 62.79±0.12 | 69.31±0.01 | 73.88±0.01 | 52.26±0.02 | 55.53±0.03 | 58.65±0.06 |
| | TimeLSTM-3 | 61.33±0.07 | 68.31±0.15 | 73.64±0.10 | 85.06±0.04 | 60.53±0.16 | 67.73±0.20 | 71.11±0.15 | 62.77±0.01 | 69.28±0.01 | 73.85±0.02 | 52.22±0.04 | 55.47±0.04 | 58.71±0.05 |
| Feat | Universal | 61.37±0.07 | 68.33±0.17 | 73.69±0.12 | 85.06±0.11 | 60.32±0.33 | 67.58±0.23 | 71.15±0.26 | 62.79±0.01 | 69.27±0.01 | 73.84±0.01 | 52.26±0.02 | 55.52±0.03 | 58.69±0.05 |
| | Current | 61.33±0.07 | 68.31±0.15 | 73.64±0.10 | 85.06±0.04 | 60.53±0.16 | 67.73±0.20 | 71.11±0.15 | 62.77±0.01 | 69.28±0.01 | 73.85±0.02 | 52.22±0.04 | 55.47±0.04 | 58.71±0.05 |
| Concat | Only LTM | 60.94±0.07 | 67.66±0.07 | 73.09±0.21 | 85.15±0.06 | 59.36±0.34 | 66.86±0.29 | 70.34±0.31 | 62.78±0.02 | 69.32±0.01 | 73.85±0.01 | 52.20±0.04 | 52.62±0.86 | 57.69±0.10 |
| | Only STM | 61.15±0.09 | 68.26±0.09 | 73.38±0.11 | 85.17±0.10 | 60.02±0.47 | 67.57±0.14 | 70.77±0.10 | 62.82±0.01 | 69.33±0.01 | 73.89±0.02 | 52.18±0.08 | 52.19±0.02 | 57.63±0.15 |
| | Both | 61.33±0.07 | 68.31±0.15 | 73.64±0.10 | 85.06±0.04 | 60.53±0.16 | 67.73±0.20 | 71.11±0.15 | 62.77±0.01 | 69.28±0.01 | 73.85±0.02 | 52.22±0.04 | 55.47±0.04 | 58.71±0.05 |
| Normalize | None | 59.14±0.08 | 65.44±0.16 | 70.53±0.10 | 85.61±0.04 | 59.97±0.11 | 67.89±0.25 | 71.22±0.21 | 63.12±0.02 | 69.65±0.01 | 74.19±0.02 | 51.46±0.01 | 54.33±0.08 | 57.39±0.08 |
| | Min Max | 57.69±0.26 | 63.33±0.20 | 68.42±0.16 | 85.23±0.05 | 59.94±0.25 | 67.76±0.15 | 71.15±0.19 | 62.37±0.05 | 69.06±0.03 | 73.61±0.06 | 51.21±0.02 | 53.10±0.04 | 55.52±0.10 |
| | Decimal Scaling | 55.03±0.19 | 60.36±0.62 | 65.72±0.35 | 85.23±0.05 | 49.99±0.01 | 50.00±0.01 | 50.00±0.00 | 50.96±1.20 | 58.53±0.87 | 63.31±0.49 | 50.00±0.00 | 50.00±0.00 | 50.00±0.02 |
| | Mean Norm | 60.23±0.13 | 66.71±0.17 | 71.86±0.08 | 84.89±0.12 | 58.03±0.57 | 65.94±0.68 | 70.07±0.14 | 63.19±0.01 | 69.68±0.02 | 74.31±0.02 | 51.86±0.01 | 54.88±0.05 | 58.11±0.02 |
| | Unit Vector | 50.01±0.04 | 54.23±0.24 | 60.02±0.48 | 85.32±0.12 | 58.12±0.25 | 65.67±0.25 | 68.97±0.17 | 56.52±0.01 | 65.43±0.02 | 71.44±0.03 | 50.75±0.02 | 52.86±0.04 | 55.53±0.11 |
| | Z-score | 61.33±0.07 | 68.31±0.15 | 73.64±0.10 | 85.06±0.04 | 60.53±0.16 | 67.73±0.20 | 71.11±0.15 | 62.77±0.01 | 69.28±0.01 | 73.85±0.02 | 52.22±0.04 | 55.47±0.04 | 58.71±0.05 |
| RNN | Vanilla RNN | 60.70±0.13 | 66.76±0.25 | 72.14±0.21 | 83.55±0.13 | 51.16±2.32 | 65.24±0.40 | 64.50±6.30 | 62.84±0.02 | 68.06±2.61 | 73.98±0.01 | 51.96±0.05 | 54.26±0.13 | 56.77±0.13 |
| | GRU | 60.34±0.19 | 66.24±0.13 | 71.23±0.21 | 81.49±0.17 | 50.96±1.44 | 60.86±0.95 | 73.38±0.11 | 62.73±0.01 | 69.16±0.01 | 73.72±0.01 | 51.80±0.04 | 54.25±0.03 | 56.79±0.06 |
| | LSTM | 58.83±0.12 | 65.12±0.14 | 69.87±0.12 | 85.18±0.08 | 58.77±0.33 | 66.52±0.22 | 70.30±0.25 | 62.30±0.01 | 68.70±0.00 | 73.25±0.02 | 51.37±0.02 | 54.05±0.07 | 57.06±0.09 |

Table 10: Comparison between Mean vs. Max aggregation operator in the packetLSTM. The mean balanced accuracy of 5 runs is reported here. We consider the synthetic datasets here and the models are run for five $p$ values, namely, 0.05, 0.25, 0.5, 0.75, and 0.95. All the results corresponding to $p$ =0.25, 0.5, and 0.75 are taken from Table 2 of the main manuscript. All the values of other parameters of packetLSTM are exactly the same, and only the aggregation operator is varied here.

| Characteristics | magic04 | | | | | a8a | | | | | SUSY | | | | | HIGGS | | | | |
|---|---|---|---|---|---|---|---|---|---|---|---|---|---|---|---|---|---|---|---|---|
| | 0.05 | 25 | 50 | 75 | 0.95 | 0.05 | 25 | 50 | 75 | 0.95 | 0.05 | 25 | 50 | 75 | 0.95 | 0.05 | 25 | 50 | 75 | 0.95 |
| Mean | 56.78 | 60.98 | 67.06 | 72.30 | 76.85 | 50.14 | 58.52 | 65.15 | 67.50 | 66.48 | 58.01 | 63.16 | 69.59 | 74.07 | 76.60 | 50.39 | 52.01 | 55.27 | 58.04 | 76.24 |
| Max | 56.74 | 61.33 | 68.31 | 73.64 | 78.13 | 50.01 | 60.53 | 67.73 | 71.11 | 73.33 | 57.57 | 62.77 | 69.28 | 73.85 | 76.57 | 50.43 | 52.22 | 55.47 | 58.71 | 77.07 |
| Mean-Max | 0.04 | -0.35 | -1.25 | -1.34 | -1.28 | 0.13 | -2.01 | -2.58 | -3.61 | -6.85 | 0.44 | 0.39 | 0.31 | 0.22 | 0.03 | -0.04 | -0.21 | -0.2 | -0.67 | -0.83 |

- Min-Max Normalization: Here, we utilize two placeholders, namely, $Mx_j^t$ and $Mn_j^t$, which represents the maximum and minimum value of feature $j$ till time $t$. The min-max normalization is then defined as $\frac{x_j^t - Mn_j^t}{Mx_j^t - Mn_j^t}$.

- Decimal Scaling: Here, all the values are scaled down by a predefined threshold as $\frac{x_j^t}{10^m}$. For all the datasets, we set $m = 3$.

- Z-score Normalization: Here, the values of each feature are normalized based on their running means ($\mu$) and standard deviation ($\sigma$) as $\frac{x_j^t - \mu_j^t}{\sigma_j^t}$, where $\mu_j^t = \mu_j^{t-} + \frac{x_j^t - \mu_j^{t-}}{k_j^t}$ and $(\sigma_j^t)^2 = \frac{v_j^t}{k_j^t - 1}$. The notation $k_j^t$ denotes the count of the feature $j$ till time $t$ and $v_j^t = v_j^{t-} + (x_j^t - \mu_j^{t-})(x_j^t - \mu_j^t)$. We utilize the above way of computing running mean and variance because of its superior numerical stability (Knuth, 1981).

- Mean Normalization: Here, the values of each feature are subtracted by their running means as $x_j^t - \mu_j^t$.

- Unit Vector Normalization: Here, we consider the whole input feature as a vector and normalize the vector as $\frac{x_j^t}{||X^t||_2}$, where $|| \cdot ||_2$ represents the $L_2$ norm.

# O   PACKETRNN AND PACKETGRU

**packetRNN**   The packetRNN framework is illustrated in Figure 5(a). Unlike the LSTM, which incorporates both short-term and long-term memory, the RNN possesses only a single memory element, known as the hidden state ($h_j^t$). Consequently, the packetRNN lacks a mechanism for integrating global information and instead maintains local information within its hidden state. These hidden states are combined using a dimension-invariant aggregation operator to generate a common hidden state for final predictions. The mathematical working of a Vanilla RNN unit within the packetRNN

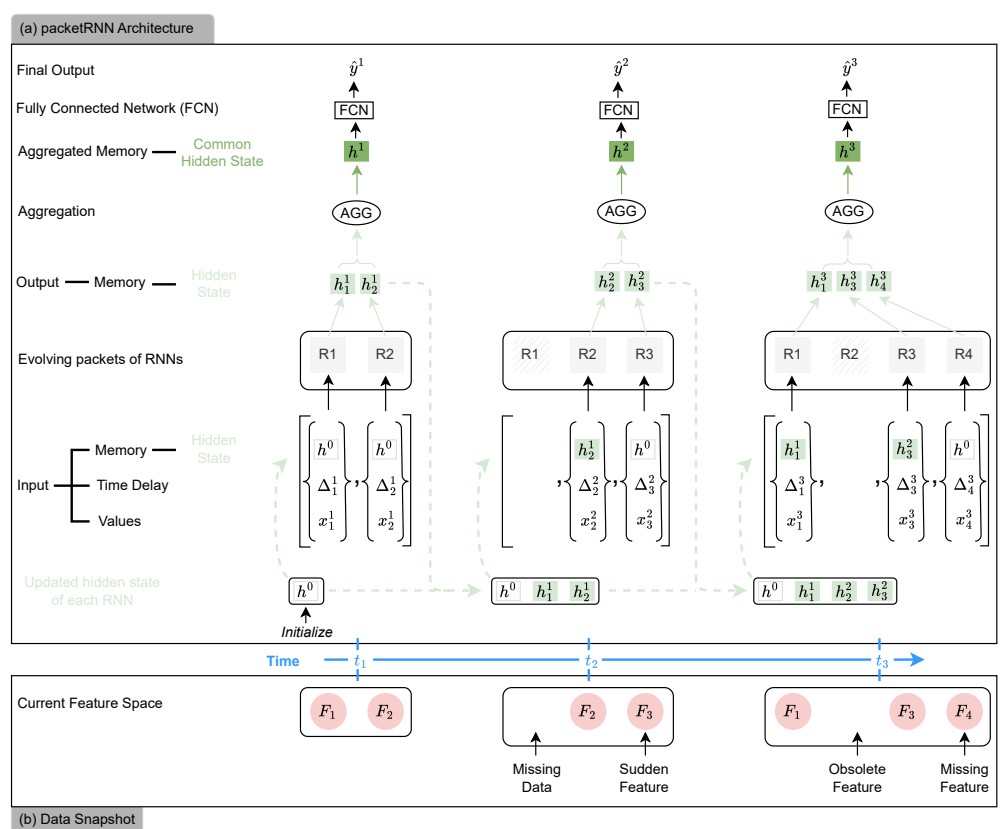

Figure 5: (a) The packetRNN architecture based on (b) the data snapshot. In this framework, the Vanilla RNN may be substituted with a GRU to establish the packetGRU framework.

framework is given by

$$h_j^t = tanh(x_j^t W_{j,xh} + h_j^{t-} W_{j,hh} + b_{j,h}), \tag{13}$$

and the final output is given by

$$\hat{y}^t = FCN(AGG(\bigcup_{j, \forall F_j \in \mathbb{F}^t} \{h_j^t\})). \tag{14}$$

**packetGRU** The vanilla RNN in Figure 5(a) can be substituted with GRU to establish the packet-GRU framework. Similar to the vanilla RNN, the GRU contains only one memory component but is enhanced with two gates: the update gate ($u_j^t$) and the reset gate ($r_j^t$). The operations of GRU within the packetGRU framework are defined by the following equations:

$$
\begin{aligned}
u_j^t &= \sigma(x_j^t W_{j,xu} + h_j^{t-} W_{j,hu} + b_{j,u}), \\
r_j^t &= \sigma(x_j^t W_{j,xr} + h_j^{t-} W_{j,hr} + b_{j,r}), \\
\tilde{h}_j^t &= tanh(x_j^t W_{j,xh} + r_j^t \odot h_j^{t-} W_{j,hh} + b_{j,h}), \\
h_j^t &= u_j^t \odot h^{t-} + (1 - u_j^t) \odot \tilde{h}_j^t,
\end{aligned}
\tag{15}
$$

and the final output of packetGRU is given by the equation 14.

## P    COMPLEXITY ANALYSIS

**Time Complexity**    The time complexity of packetLSTM is $\sum_{t=1}^{T} O(g(|\mathbb{F}^t|) * s^2 + P)$, where $T$ is the number of instances, $O(g(|\mathbb{F}^t|) * s^2)$ is the time complexity to process all the activated LSTMs at time $t$ corresponding to $\mathbb{F}^t$ features, and $O(P)$ broadly denotes the constant time required to perform other fixed operations like aggregations and final prediction. We utilize the torch.matmul() function for matrix multiplication of LSTMs. The time required for each LSTM is dependent on its hidden size ($s$) and performs 3 matrix multiplication of size $(3s, s + 1)$ and $(s + 1, 1)$ requiring a time complexity of $O(s^2)$. However, we vectorize the operation of $|\mathbb{F}^t|$ LSTMs by single matrix multiplication of $(|\mathbb{F}^t|, 3s, s + 1)$ and $(|\mathbb{F}^t|, s + 1, 1)$. Note that the time required by torch.matmul() does not scale linearly with $|\mathbb{F}^t|$; rather, it just takes a small overhead depending on the type of hardware and other dependencies. Here, we denote this overhead as a function of $|\mathbb{F}^t|$ as $g(|\mathbb{F}^t|)$ where $1 \leq g(|\mathbb{F}^t|) \leq |\mathbb{F}^t|$. This is further corroborated by the time required by packetLSTMs on each synthetic dataset with different $p$ values (see Table 8). For example, the time required to process the whole HIGGS dataset takes 4396.83 and 4500.17 seconds for $p = 0.25$ and 0.5, respectively. Even though the value of $|\mathbb{F}^t|$ doubles from $p = 0.25$ to 0.5, the time required doesn't increase by the same factor. Therefore, the time required by the $|\mathbb{F}^t|$ LSTMs at time $t$ would be $O(g(|\mathbb{F}^t|) * s^2)$. Finally, since it is an online learning task and each instance is processed sequentially, the total time complexity of $T$ instances would be $\sum_{t=1}^{T} O(g(|\mathbb{F}^t|) * s^2 + P)$. It is difficult to find the exact form of the function $g$. However, based on the time required by packetLSTM on each dataset with increasing $p$, it can be safely assumed that the value of $g(|\mathbb{F}^t|)$ is closer to 1 than $|\mathbb{F}^t|$. Therefore, the packetLSTM model demonstrates scalability in terms of time complexity corresponding to the number of features.

**Space Complexity**    The space complexity of packetLSTM is directly dependent on the space complexity of an LSTM. Let us denote the space complexity of an LSTM by $O(L)$. At each time $t$, we have a universal feature space of $|\bar{\mathbb{F}}^t|$ cardinality, therefore, corresponding $|\bar{\mathbb{F}}^t|$ LSTMs are present in the packetLSTMs. The space complexity of the final prediction network and aggregation function are fixed and denoted by $O(K)$. Therefore, the total space complexity of packetLSTM at time $t$ can be given by $O(|\bar{\mathbb{F}}^t| * L + K)$. Note that the space complexity of packetLSTM increases with the arrival of sudden and missing features. Therefore, the limitation of packetLSTM is that it is not scalable in terms of space complexity corresponding to the feature size. However, it is noteworthy that packetLSTM effectively manages up to 7500 features, as demonstrated with the imdb dataset. Additionally, we present a strategy to further mitigate space complexity limitation in section Q of the Appendix.

**Number of Parameters**    Here, we provide the number of learnable parameters in the packetLSTM framework with Time-LSTM 3 as the time modeling unit. The mathematical formulation is given by equations 2-8. The input gate ($i$), time gate 1 ($T1$), time gate 2 ($T2$), cell state ($\tilde{c}$), long-term memory ($c$), and output gate ($o$) requires $s^2 + 3s$, $3s$, $3s$, $s^2 + 2s$, $s^2 + 2s$, and $s^2 + 4s$ learnable parameters, respectively. Therefore, the total number of learnable parameters in an LSTM unit is $4s^2 + 17s$. The fully connected layer accounts for $2s^2 + 4s + 2$ parameters. Therefore, the total number of learnable parameters at each instance is $|\mathbb{F}^t| * (4s^2 + 17s) + 2s^2 + 4s + 2$. For all the experiments, the value of $s$ is 64. In this article, the maximum number of parameters for each dataset would correspond to the worst-case scenario where all the features are present, resulting in a maximum of $\sim$183K, $\sim$131M, $\sim$2M, $\sim$148K, and $\sim$375K learnable parameters for magic04, imdb, a8a, SUSY, and HIGGS, respectively. Consequently, packetLSTM requires 1.40 MB, 999.45 MB, 15.26 MB, 1.13 MB, and 2.85 MB memory with 64-bit precision for magic04, imdb, a8a, SUSY, and HIGGS, respectively. Considering that a small large language model of 1 billion parameters is common nowadays, packetLSTM can handle $\sim$57K features with 1 billion parameters. We believe that a practical application would have a number of features way less than $\sim$57K.

## Q    DROPPING FEATURES TO RESOLVE SPACE COMPLEXITY

The space complexity increases with the number of features, as discussed above. However, packetLSTM easily handles even the 7500 features in the imdb dataset. The total number of learnable parameters of packetLSTM for the imdb dataset is $\sim$131M. Therefore, packetLSTM with 1B param-

eters and a hidden size of 64 can handle around ∼57K features. Hence, we argue that packetLSTM can deal with high-dimensional data. However, we also propose a solution to curb the space complexity by defining a maximum limit (say $l_f$) on the number of LSTMs. When the number of features in the universal feature space $|\bar{\mathbb{F}}^t| > l_f$, we drop $|\bar{\mathbb{F}}^t| - l_f$ features. Here, dropping the feature means that the corresponding LSTM is reinitialized and assigned to some new features that arrived at time $t$. The dropped feature can come in future instances ($> t$). However, this feature will then be considered as a sudden feature. We employ KL-Divergence to determine the dropped features. After processing instance $t - 1$, packetLSTM has the short-term memory of each LSTM and the common long-term memory. The KL divergence between each short-term and common long-term memory is determined. The feature corresponding to the short-term memory, which has the lowest KL-Divergence, is dropped. This is because the common long-term memory already holds the dropped feature's short-term memory information and is used for final prediction. We also put a limit on the number of times a feature is seen ($i_l$) by packetLSTM before it can be considered for dropping. We experimented with the imdb dataset since it has the highest number of features (7500). The best hyperparameters found for packetLSTM on the imdb dataset in Table 7 are used here. The $l_f$ and $i_l$ are set to 100 and achieved a balanced accuracy of 78.92, which still performs better than 7 out of 10 baseline models (see Table 1 in main manuscript). The 78.92 balanced accuracy is lower than the packetLSTM without any limits (85.06). So, there is a tradeoff between performance and the space complexity.

## R  SINGLE LSTM

The three employed imputation techniques are:

- Forward fill: The missing value of a feature is imputed with its last observed value.

- Mean: The missing value of a feature is imputed with its forward mean determined in a rolling manner. That is, the last $K$ observed values of a feature are considered to calculate its mean. Here $K = 5$.

- Gaussian Copula: (Zhao et al., 2022) proposed using Gaussian copula for online streaming data with a known $N$, where $N$ is the total number of features. This method argues that data points are generated from a latent Gaussian vector, which is then transformed to match the marginal distributions of observed features. However, Gaussian Copula requires storing a matrix of $K$ instances to perform imputation. The method fails if the matrix's determinant is 0 or if a feature's values within the matrix are identical. The magic04 and SUSY require $K = 5$, and HIGGS needs $K = 30$. The method does not work for a8a and imdb for even $K = 300$, excluding its application to these datasets.

The hyperparameter search of a single LSTM is performed similarly to packetLSTM. The hyperparameters of the single LSTM model are hidden size, number of layers, and learning rate. We searched for the best hidden size among 32, 64, 128, and 256. We determined the optimal number of layers between 1, 2, 3, and 4. Similar to packetLSTM, we tested a range of learning rates (0.001, 0.0005, 0.0001, 0.00005). We further searched in the vicinity of the optimal learning rate found in the previous step. The optimal hidden size and number of layers for each dataset are found to be 32 and 1, respectively. The best learning rate is 0.001, 0.0006, 0.0001, 0.0002, and 0.0008 for magic04, a8a, SUSY, HIGGS, and imdb, respectively. Similar to packetLSTM, we employed Z-score streaming normalization.

## S  CHALLENGING SCENARIOS ON HIGGS AND SUSY

### S.1  HIGGS

Here, we provide the exact value of the balanced accuracy used in Figure 3 and 4 from section 7 of the main manuscript. Table 11 provides the results associated with the experiments on sudden, obsolete, and reappearing features on the HIGGS dataset. Additionally, Table 12 details the comparative results of packetLSTM versus packetLSTM retrained across five data intervals.

Table 11: The mean±standard deviation of the balanced accuracy of sudden, obsolete, and reappearing experiments conducted in section 7 of the main manuscript. This table reflects the values of the results presented in Figure 3 and 4 and corresponds to the HIGGS dataset.

| Data Interval | OLVF | OLIFL | OVFM | Aux-Drop | **packetLSTM** |
|---|---|---|---|---|---|
| | | | **Sudden** | | |
| First | 51.09 | 51.07 | 52.96±0.00 | 53.51±0.05 | **54.30±0.04** |
| Second | 54.45 | 53.10 | 54.49±0.00 | 55.77±0.04 | **56.98±0.03** |
| Third | 54.14 | 54.68 | 54.53±0.00 | 57.98±0.10 | **60.20±0.13** |
| Fourth | 54.08 | 54.93 | 54.82±0.00 | 60.09±0.16 | **62.37±0.12** |
| Fifth | 53.86 | 55.15 | 55.31±0.00 | 61.24±0.20 | **63.28±0.06** |
| | | | **Obsolete** | | |
| First | 53.56 | 52.32 | 54.62±0.01 | 57.64±0.18 | **59.90±0.08** |
| Second | 51.80 | 52.27 | 52.21±0.01 | 55.48±0.16 | **58.15±0.09** |
| Third | 51.70 | 51.94 | 52.16±0.01 | 55.69±0.11 | **57.01±0.03** |
| Fourth | 51.48 | 51.80 | 51.41±0.01 | 52.38±0.09 | **53.77±0.05** |
| Fifth | 50.59 | 50.40 | 50.73±0.00 | 50.93±0.07 | **51.32±0.02** |
| | | | **Reappearing** | | |
| First | 51.42 | 53.34 | 53.03±0.00 | 53.52±0.04 | **58.74±0.08** |
| Second | 52.07 | 51.39 | 52.58±0.01 | 53.20±0.02 | **53.27±0.07** |
| Third | 50.60 | 54.78 | 50.79±0.00 | 52.98±0.19 | **60.42±0.11** |
| Fourth | 51.30 | 51.88 | 51.28±0.01 | 51.65±0.05 | **53.94±0.08** |
| Fifth | 50.63 | 55.27 | 50.78±0.01 | 50.79±0.09 | **60.54±0.13** |

Table 12: The exact values represented in the experiment 'Learning Without Forgetting' in Figure 4 from section 7 of the main manuscript. This result corresponds to the HIGGS dataset.

| Model | First | Second | Third | Fourth | Fifth |
|---|---|---|---|---|---|
| packetLSTM | 58.74±0.08 | **53.27±0.07** | **60.42±0.11** | **53.94±0.08** | **60.54±0.13** |
| packetLSTM-Retraining | 58.74±0.08 | 53.22±0.06 | 58.81±0.05 | 53.01±0.06 | 58.65±0.07 |

The reappearing experiment in the main manuscript and Figure 4 shows that packetLSTM mitigates catastrophic forgetting. We further justify this by comparing packetLSTM with Aux-Drop (the best baseline method as shown in Table 1).

**Aux-Drop:** The performance of Aux-Drop decreases from 53.52 in the first interval to 52.98 in the third interval. The performance further decreases to 50.93 in the fifth interval. Aux-Drop also shows a performance decline from 55.48 in the second interval to 52.38 in the fourth interval. Therefore, Aux-Drop suffers from catastrophic forgetting.

**packetLSTM:** The performance of packetLSTM increases from 58.74 in the first interval to 60.42 in the third interval. The performance further increases to 60.54 in the fifth interval. The packetLSTM also shows a performance increase from 53.27 in the second interval to 53.94 in the fourth interval. Thus, packetLSTM mitigates catastrophic forgetting.

It can be argued that the performance increase of packetLSTM in the third interval is not due to the mitigation of catastrophic forgetting; it is instead due to a likely situation of better predictive capability of the features present in the third interval compared to the first interval.

To refute the above argument, we retrained the packetLSTM in each interval – a new initialized packetLSTM was considered for each interval – and referred to it as packetLSTM-Retraining (see Figure 4 and Table 12). The performance of the packetLSTM (60.42) outperforms packetLSTM-Retraining (58.81) in the third interval, which shows that packetLSTM can retain its learning from the first interval. Similar performance increase is observed in the fourth and fifth intervals.

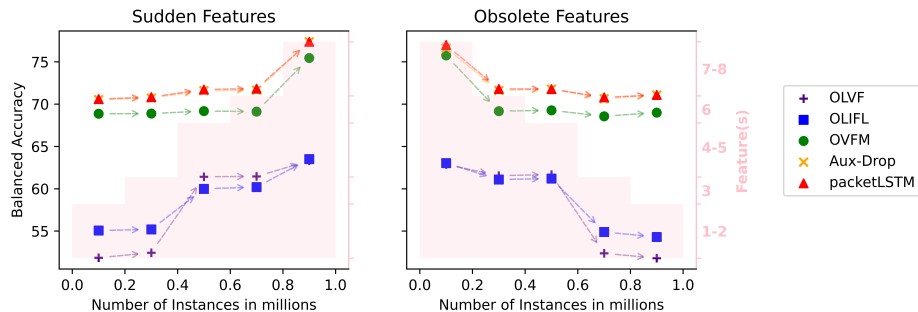

Figure 6: Model performance in the scenario of sudden and obsolete features. The mean balanced accuracy in each data interval (e.g., 0 - 0.2 M) is shown in its middle (0.1 M). The pink-colored y-axis represents the Feature(s). For e.g., '1-2' means features 1 to 2 are available in instances shaded with pink color. Both y-axes are shared by the two graphs. This graph corresponds to the SUSY dataset.

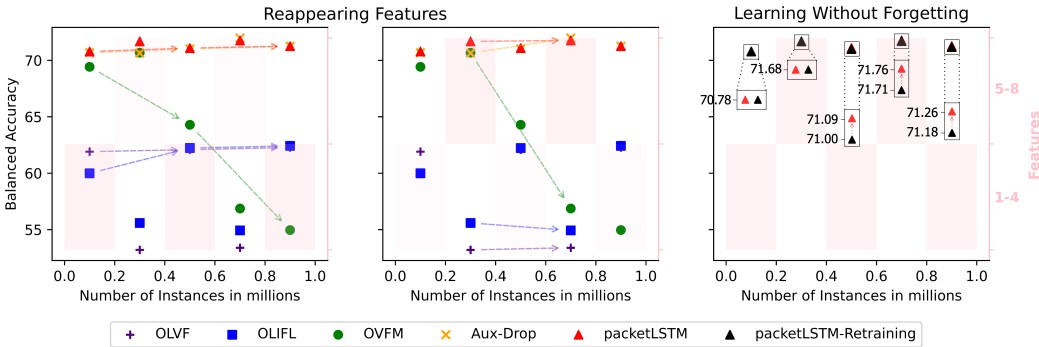

Figure 7: Model performance in the scenario of reappearing features. The mean balanced accuracy in each data interval (e.g., 0 - 0.2 M) is shown in its middle (0.1 M). The pink-colored y-axis represents the Features. For e.g., '1-4' means features 1 to 4 are available in instances shaded with pink color. Both y-axes are shared by the two graphs. This graph corresponds to the SUSY dataset.

## S.2 SUSY

We perform three challenging experiments – sudden features, obsolete features, and reappearing features – on the SUSY dataset, replicating the experiments conducted on the HIGGS dataset. The data creation settings for the SUSY dataset are identical to those employed for the HIGGS dataset. We considered the OLVF, OLIFL, OVFM, and Aux-Drop baselines for comparative analysis. The performance of each method is illustrated in Figures 6 and 7, with precise values detailed in Tables 13 and 14.

**Sudden Features**  Similar to the approach used for the HIGGS dataset, 20% of the features are sequentially added at each interval in the SUSY dataset, with values rounded to the nearest integer. Given that SUSY comprises 8 features, the initial data interval includes features 1 and 2, the subsequent interval contains features 1 through 3, and this incremental addition continues as depicted in the leftmost graph of Figure 6. All models exhibit a pattern of performance improvement as sudden features are introduced at each interval, as evident in Figure 6. Notably, the packetLSTM consistently outperforms all baselines in each data interval, followed by Aux-Drop, as detailed in Table 13. This underscores the efficacy of packetLSTM in handling sudden features.

**Obsolete Features**  The data arrival pattern in this experiment is the inverse of that observed in the sudden feature experiment, as illustrated in the middle graph of Figure 6. Generally, the performance

Table 13: The mean±standard deviation of the balanced accuracy of sudden, obsolete, and reappearing experiments presented in Figure 6 and 7 and corresponds to the SUSY dataset.

| Data Interval | OLVF | OLIFL | OVFM | Aux-Drop | packetLSTM |
|---|---|---|---|---|---|
| First | 51.85 | 55.06 | 68.86±0.00 | 70.53±0.04 | **70.60±0.03** |
| Second | 52.44 | 55.20 | 68.88±0.00 | 70.72±0.07 | **70.83±0.02** |
| Third | 61.42 | 59.99 | 69.18±0.00 | 71.59±0.03 | **71.73±0.02** |
| Fourth | 61.46 | 60.20 | 69.12±0.00 | 71.64±0.05 | **71.83±0.03** |
| Fifth | 63.30 | 63.51 | 75.45±0.00 | **77.37±0.08** | **77.37±0.05** |
| **Obsolete** | | | | | |
| First | 62.87 | 63.03 | 75.75±0.00 | 76.62±0.06 | **76.99±0.03** |
| Second | 61.55 | 61.09 | 69.17±0.00 | 71.61±0.11 | **71.80±0.06** |
| Third | 61.67 | 61.20 | 69.27±0.00 | **71.79±0.06** | 71.78±0.01 |
| Fourth | 52.37 | 54.89 | 68.56±0.00 | 70.64±0.06 | **70.80±0.02** |
| Fifth | 51.80 | 54.30 | 69.00±0.00 | **71.08±0.04** | **71.09±0.02** |
| **Reappearing** | | | | | |
| First | 61.91 | 59.99 | 69.43±0.00 | 70.67±0.02 | **70.78±0.03** |
| Second | 53.20 | 55.59 | 70.68±0.00 | 70.64±0.78 | **71.68±0.03** |
| Third | 62.08 | 62.23 | 64.29±0.03 | 71.01±0.04 | **71.09±0.02** |
| Fourth | 53.39 | 54.93 | 56.87±0.62 | **71.98±0.04** | 71.76±0.03 |
| Fifth | 62.28 | 62.42 | 54.96±1.36 | **71.26±0.05** | **71.26±0.01** |

Table 14: The exact values represented in the experiment 'Learning Without Forgetting' in the rightmost graph of Figure 7. This result corresponds to the SUSY dataset.

| Model | First | Second | Third | Fourth | Fifth |
|---|---|---|---|---|---|
| packetLSTM | 70.78±0.03 | 71.68±0.03 | **71.09±0.02** | **71.76±0.03** | **71.26±0.01** |
| packetLSTM-Retraining | 70.78±0.03 | 71.68±0.03 | 71.00±0.02 | 71.71±0.02 | 71.18±0.03 |

of all models declines as features become obsolete in each subsequent interval. Interestingly, an increase in performance from the fourth to the fifth interval is observed in models such as OVFM, Aux-Drop, and packetLSTM. A plausible explanation for this phenomenon may relate to the nature of the SUSY data, where the exclusion of feature 6 notably impacts performance more severely in the fourth interval. The packetLSTM consistently outperforms other models throughout each data interval, followed by Aux-Drop, demonstrating its robust capability in handling obsolete features effectively.

**Reappearing** The performance of packetLSTM, Aux-Drop, and OLVF improves when previously encountered sets of features reappear, as depicted in Figure 7. Furthermore, packetLSTM exhibits performance gains of 0.09, 0.05, and 0.08 in the third, fourth, and fifth intervals, respectively, compared to its retrained counterpart, as shown in the rightmost graph of Figure 7 and Table 14. This enhancement reaffirms the 'learning without forgetting' capability of packetLSTM. Notably, packetLSTM is the sole method that consistently demonstrates the 'learning without forgetting' capability across both the HIGGS and SUSY datasets. Moreover, packetLSTM consistently gives the best result in four out of five data intervals, followed by Aux-Drop, as detailed in Table 13, underscoring its superior performance.

# T  CATASTROPHIC FORGETTING IN ONLINE LEARNING VERSUS ONLINE CONTINUAL LEARNING

In this article, we present the 'learning without forgetting' capability of packetLSTM. The 'learning without forgetting' capability is also referred to as mitigating catastrophic forgetting in the existing literature (Hoi et al., 2021). We explore this capability in an online learning setting, where the model receives and processes instances sequentially without any buffer storage.

The problem of catastrophic forgetting is extensively studied in a parallel and widely recognized field of online continual learning (Li & Hoiem, 2017; Wang et al., 2024). Online continual learning is the field of machine learning where a model continually learns new tasks. In this scenario, tasks are delivered sequentially, yet all the instances for each task are presented at once. Consequently, the learning for each task is conducted in an offline setting, utilizing all data related to a task before transitioning to the subsequent one. For example, a model may initially learn from data related to task 1 in an offline batch mode. Subsequently, it proceeds to learn from data pertaining to task 2 in a similar batch mode, and this pattern continues. The introduction of new tasks may result in the addition of new classes, a shift in data distribution, or the arrival of an entirely new task domain. In the context of online continual learning, catastrophic forgetting refers to the challenge a model faces in retaining its ability to perform previously learned tasks.

Although online continual learning shares similarities with online learning regarding sequential task learning, it differs significantly in its execution within individual tasks. In online continual learning, the model adheres to a batch training paradigm, which diverges from the principles of online learning algorithms. Consequently, the phenomenon of catastrophic forgetting, as observed in online learning, distinctly differs from that in online continual learning. It is worth noting that the packetLSTM framework, with certain modifications, may potentially be adapted for use in online continual learning to address catastrophic forgetting. However, this application extends beyond the scope of our article.

## U  TRANSFORMER

### U.1  PADDING

The two input padding methods are:

- *Only Values*: For each dataset, zeros are added to the sequence following the available feature values until $f_l$, where $f_l = N$, where $N$ is the total number of features. Note that this contradicts the sixth characteristic of haphazard inputs. However, to compare packetLSTM with Transformer, we assume that $N$ is known. The specific $N$ for each dataset is defined in the column labeled '#Features' in Table 5 of the Appendix. An example of an input instance at time $t$ where feature 1 and $j$ is available is $[x_1^t, x_j^t, 0, ..., 0]$.
- *Pairs*: Each available feature value is paired with its corresponding feature ID, and the sequence of these pairs is padded with zeros till $f_l$. The feature ID ranges from 1 to $N$, and $f_l = 2N$. An Example is $[x_1^t, 1, x_j^t, j, 0, ..., 0]$

Padded inputs are initially processed through a linear embedding layer, which outputs embedding of dimension $d$. The input dimension for this layer is $N$ and $2N$ for *Only Values* and *Pairs*, respectively. The embedding is then passed to an encoder. The encoder consists of $n_l$ encoder layers. Each encoder layer comprises $n_h$ heads and maintains dimensions of size $d$ for both input and output. The encoder's output feeds into the same fully connected neural network utilized in packetLSTM, consisting of a linear layer of dimensions $(d, d)$, a ReLU activation, and another linear layer leading to the output classes, culminating in a softmax layer for predictions.

The hyperparameters search is performed sequentially, as in the packetLSTM. We searched the optimal value of $d$ among 32, 64, 128, 256, and 512, $n_h$ among 1, 2, 4, 8, and 16, $n_l$ among 1, 2, 3, and 4. Similar to packetLSTM, we tested a range of learning rates (0.001, 0.0005, 0.0001, 0.00005). We further searched in the vicinity of the optimal learning rate found in the previous step. Optimal hyperparameters include a $d$ of 128 for magic04 and 32 for other datasets, $n_h$ of 4 for SUSY and imdb, 8 for magic04, and 16 for a8a and HIGGS. The $n_l$ is 1 for all the datasets, with the learning rate of 0.0002 for magic04 and a8a and 0.0001 for SUSY, HIGGS, and imdb. The Z-score is employed for streaming normalization.

### U.2  NATURAL LANGUAGE

We experimented with two models, DistilBERT (Sanh et al., 2019) and BERT (Devlin et al., 2019), to process haphazard inputs. The inputs to the model are created in two ways:

- *Values*: Sequences of available feature values, formatted as a string (example "$x_1^t$ $x_j^t$").

Table 15: Performance of DistilBERT and BERT on haphazard inputs. The synthetic dataset is considered for $p = 0.5$, and the performance of 3 runs is reported here. Since the mean balanced accuracy is around 50 in all cases, we did not conduct further experiments.

| Model | Data Type | Metric | magic04 | imdb | a8a | SUSY | HIGGS |
|-------|-----------|--------|---------|------|-----|------|-------|
| DistilBERT | Values | bAcc | 50.89±1.27 | 50.48±0.06 | 50.0±0.00 | 50.02±0.04 | 50.0±0.00 |
| | | Acc | 65.21±0.57 | 50.48±0.06 | 75.92±0.0 | 54.21±0.02 | 52.95±0.0 |
| | | ROC | 50.69±2.02 | 50.08±0.03 | 49.64±0.03 | 50.13±0.08 | 50.05±0.01 |
| | | PRC | 65.17±1.26 | 50.09±0.08 | 83.01±0.19 | 45.94±0.14 | 52.98±0.01 |
| | | Err | 6616.67±107.95 | 12379.0±15.75 | 7841.67±0.47 | 457861.67±236.68 | 470490.67±22.9 |
| | | Time | 487.88±0.65 | 705.84±3.61 | 866.5±3.32 | 19767.84±648.36 | 20385.68±2652.15 |
| | Input Pairs | bAcc | 50.01±0.02 | 50.35±0.06 | 50.0±0.00 | - | - |
| | | Acc | 55.41±13.31 | 50.35±0.06 | 75.92±0.0 | | |
| | | ROC | 50.01±0.07 | 50.12±0.03 | 49.62±0.01 | | |
| | | PRC | 75.83±7.69 | 50.06±0.02 | 83.01±0.06 | | |
| | | Err | 8481.67±2530.97 | 12412.33±14.38 | 7841.67±0.47 | | |
| | | Time | 496.47±3.26 | 846.39±8.61 | 938.85±2.88 | | |
| BERT | Values | bAcc | 50.0±0.01 | 50.26±0.02 | 50.0±0.00 | - | - |
| | | Acc | 64.83±0.0 | 50.26±0.22 | 75.92±0.0 | | |
| | | ROC | 50.0±0.05 | 50.04±0.02 | 49.67±0.06 | | |
| | | PRC | 80.71±1.03 | 50.02±0.01 | 82.87±0.02 | | |
| | | Err | 6689.0±0.82 | 12434.0±55.37 | 7842.0±0.0 | | |
| | | Time | 619.53±10.21 | 1137.5±3.14 | 1470.69±6.11 | | |
| | Input Pairs | bAcc | 50.0±0.00 | 50.44±0.02 | 50.0±0.00 | - | - |
| | | Acc | 64.82±0.01 | 50.44±0.02 | 75.92±0.0 | | |
| | | ROC | 49.93±0.03 | 50.02±0.06 | 49.62±0.01 | | |
| | | PRC | 80.91±1.15 | 50.07±0.04 | 82.95±0.11 | | |
| | | Err | 6691.33±1.25 | 12390.33±4.64 | 7842.0±0.0 | | |
| | | Time | 631.61±2.44 | 1006.53±5.06 | 1205.76±4.31 | | |

- *Input Pairs*: Sequences where each feature value is paired with its feature ID (example "$[x_1^t, 1]\ [x_j^t, j]$").

We utilize the default hidden size of both DistilBERT and BERT models, and learning rates are determined by hyperparameter search among 0.001, 0.0005, 0.0001, and 0.00005. The balanced accuracy on the imdb and synthetic datasets (with $p = 0.5$) is around 50 for both models (see Table 15), with the highest being 50.89 for DistilBERT using *Values* on the magic04 dataset. Given that the models did not learn to classify haphazard inputs and considering the extensive computation time required for the BERT models, we did not conduct further experiments.

## U.3    SET TRANSFORMER

We utilize the encoder and decoder provided in the Set Transformer article (Lee et al., 2019), which is permutation invariant and handles variable length inputs. The hyperparameters are hidden size, number of heads, number of induction (inducing points), and learning rate. Please refer to Set Transformer (Lee et al., 2019) for more details about the architecture.

We conduct a hyperparameter search similar to packetLSTM. We searched the optimal value of hidden size and the number of induction points among 32, 64, 128, 256, and 512, and the number of heads among 1, 2, 4, and 8. Similar to packetLSTM, we tested a range of learning rates (0.001, 0.0005, 0.0001, 0.00005). We further searched in the vicinity of the optimal learning rate found in the previous step. The optimal hidden sizes are 64 for magic04 and 32 for the rest of the dataset. The best value of the number of induction points is 64 for magic04 and HIGGS, 32 for imdb and a8a, and 256 for SUSY. The optimal number of layers is 1 for magic04 and HIGGS, 2 for imdb and a8a, and 8 for SUSY. The best learning rate is 0.0001, 0.00007, 0.0002, 0.00008, and 0.00006 for magic04, imdb, a8a, SUSY, and HIGGS, respectively.

## U.4    HAPTRANSFORMER

We create embeddings of dimension ($d$) for each feature, initialized with He initialization (He et al., 2015), and designed to be learnable to capture feature representations effectively. At each time instance, only embeddings for available features are utilized and are collectively denoted as $E_t$. Given that the shape of $E_t$ would be $(1, |\mathbb{F}^t|, d)$, it varies in length due to the changing number of features at time $t$ ($|\mathbb{F}^t|$). Therefore, the decoder of the Set Transformer (Lee et al., 2019), which handles variable length inputs, is utilized to process $E_t$. The decoder accepts an input of size $d$ for

Table 16: Performance of packetLSTM with complete features ($p = 1$) on synthetic datasets. All the metrics are reported as mean $\pm$ standard deviation of five runs. The bAcc, Acc, ROC, PRC, Err, and Time stand for balanced accuracy, accuracy, AUROC, AUPRC, number of errors, and execution time, respectively.

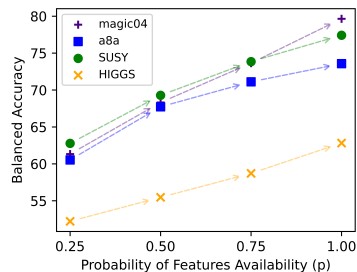

Figure 8: Performance of packetL-STM on various probability of features availability.

| Metric | magic04 | a8a | SUSY | HIGGS |
|---|---|---|---|---|
| bAcc | 79.65±0.13 | 73.57±0.15 | 77.42±0.05 | 62.66±0.18 |
| Acc | 83.68±0.12 | 83.37±0.08 | 78.11±0.05 | 62.98±0.19 |
| ROC | 87.79±0.06 | 87.54±0.11 | 84.92±0.05 | 67.95±0.28 |
| PRC | 90.49±0.11 | 95.49±0.06 | 85.17±0.06 | 69.60±0.24 |
| Err | 3103.2±21.93 | 5415.4±26.19 | 218943.6±492.38 | 370198±1863.46 |
| Time | 79.02±8.97 | 261.74±14.66 | 4146.19±98.9 | 5589.51±94.65 |

each feature and includes $n_h$ number of heads. The output from the decoder is then passed through a softmax layer to generate predictions.

We conduct a hyperparameter search for $d$, $n_h$, and learning rate following the methodology used for packetLSTM. We determined the optimal $d$ among 32, 64, 128, 256, and 512, and $n_h$ among 1, 2, 4, and 8. The best learning rate was determined among 0.001, 0.0005, 0.0001, and 0.00005. We further searched in the vicinity of the optimal learning rate found in the previous step. Optimal values identified include a $d$ of 64, 512, 256, 32, and 64, $n_h$ of 2, 1, 2, 2, and 4 for magic04, imdb, a8a, SUSY, and HIGGS, respectively. The learning rate is found to be 0.0005, 0.00007, 0.00009, 0.00009, and 0.00008 for magic04, imdb, a8a, SUSY, and HIGGS, respectively.

## V HAPHAZARD INPUTS VS OTHER FIELDS OF VARYING FEATURE SPACE

The field of varying feature space is also studied as feature evolvable streams (Hou et al., 2017; Zhang et al., 2020) and incremental and decremental features (Hou & Zhou, 2018; Dong et al., 2021). However, both these fields assume some form of structure in their data stream, which contradicts the characteristics of haphazard inputs. Specifically, the feature evolvable streams work in batches and assume that there is an overlap period between the transition where old features vanish and new features occur. Similarly, incremental and decremental features assume that the data arrives in batches, where the initial batch consists of both vanishing and surviving features and subsequent batches encompass surviving and newly augmented features. The assumption of the batch and the structure of the data within the batch limits the applicability of both fields in haphazard inputs.

## W PACKETLSTM WITH COMPLETE FEATURES

We set $p = 1$ in the synthetic datasets to determine the upper bound performance of packetLSTM when all the features are present at each time instance. The corresponding results are provided in Table 16. We present a comparison of packetLSTM's performance across varying $p$ values (0.25, 0.5, 0.75, and 1) in Figure 8. Unsurprisingly, the performance of packetLSTM increases with decreasing haphazardness (i.e., increasing $p$ value) in the data. The most significant decline in performance occurs at $p = 0.25$, which is expected due to the increased haphazardness.

