# OpenReview forum: "packetLSTM: Dynamic LSTM Framework for Streaming Data with Varying Feature Space"
_ICLR.cc/2025/Conference — Submitted to ICLR 2025_

### Official Review · Reviewer_5PVp · 2024-11-01

**Soundness:** 3
**Presentation:** 2
**Contribution:** 2
**Rating:** 5
**Confidence:** 4

**Summary:**

This paper introduces packetLSTM, a novel LSTM-based framework tailored for handling streaming data with varying input feature dimensions. Each feature in the streaming data is managed by an individual LSTM unit, promoting effective handling of temporal variations in feature space. The packetLSTM architecture is claimed to mitigate catastrophic forgetting by leveraging both feature-specific short-term memory and a shared global memory. The model shows competitive or superior performance compared to baselines on multiple datasets and is versatile across other RNN architectures.

**Strengths:**

1. The architecture's capacity to accommodate missing features through individual LSTM management enhances its robustness in varied real-world data.
2. The inclusion of ablation studies and detailed results across various dataset conditions strengthens the reliability of the model's claims.
3. Comprehensive experiments on five datasets, along with competitive metrics, solidifies the framework’s potential as a state-of-the-art solution.

**Weaknesses:**

1. Given that available features trigger the activation of corresponding LSTMs, what are the hidden states from the previous timestamp that are input into the currently activated LSTM but not activated in the last timestamp?

2. The proposed method demonstrates limited novelty, as it directly adapts LSTM to handle dimension-varying streams.

3. In Table 2, the authors indicate that the Max operator yields the best performance as the aggregation (AGG) function. However, this approach may overlook valuable information from other dimensions. Including an ablation study with a self-attention mechanism in the AGG operator, which may potentially enhance the aggregation of information across different features.

4. The paper would benefit from a more thorough theoretical analysis.

5. The yellow color text in Figure 2 is too light, making it difficult to read clearly.

**Questions:**

see weakness

---

> ### Author Response · Authors · 2024-11-16
> **Thanking reviewer for their feedback**
>
> We thank the reviewer for their constructive feedback. We appreciate that the reviewer recognized the robustness of our method and extensive evaluation performed to solidify our framework as a state-of-the-art solution
>
> We provide answers to all the reviewer's comments below and reflect them in our updated manuscript in blue color wherever applicable. We hope we were able to provide satisfactory answers. Please feel free to ask if the reviewer requires any further clarification.

---

> > ### Author Response · Authors · 2024-11-16
> > **Reply to Weakness 1**
> >
> > **W.1.** Given that available features trigger the activation of corresponding LSTMs, what are the hidden states from the previous timestamp that are input into the currently activated LSTM but not activated in the last timestamp?
> >
> > **Answer:** We would like to draw the attention of the reviewer to Figure 2, where the LSTM L1 (or $L_1$) corresponding to feature $F_1$ is activated at time $t_3$ and $t_1$ but not activated at time $t_2$. The hidden states that are input to L1 at time $t_3$ are the common long-term memory ($c^2$) and the short-term memory ($h_1^1$). The $c^2$ is determined at time $t_2$, and $h_1^1$ is the short-term memory of feature $F_1$ from time $t_1$. This is also mentioned in lines 155-158 as stated below:
> >
> > > *Each LSTM ($L_j$) receives inputs comprising the feature value ($x_j^t$), time delay ($\Delta_j^t$), its previous short-term memory ($h_j^{t_-}$), and common long-term memory ($c^{t-1}$), as labeled \textit{Input} in Figure 2(a). The $\Delta_{j}^{t}$ measures the time difference between the current time $t$ and the last observed time $t_-$ for feature $j$.*

---

> > ### Author Response · Authors · 2024-11-16
> > **Reply to Weakness 2**
> >
> > **W.2.** The proposed method demonstrates limited novelty, as it directly adapts LSTM to handle dimension-varying streams.
> >
> > **Answer:** We gently beg to differ. While the reviewer is correct that we adapt LSTM, we note that LSTM cannot be used DIRECTLY for haphazard inputs as the number of input features can change at each time step, and the total number of features is unknown. On the other hand, packetLSTM "dynamically handles varying input dimensions through the activation/deactivation of LSTMs," as noted by reviewers 1 and 2.
> >
> > The main obstacle in adapting LSTM directly is that LSTM requires a fixed feature space, which necessitates the availability of all features. This requirement can be met through imputation, although it results in contradicting the sixth characteristic of haphazard inputs where the total number of features is unknown. We have discussed this in the paragraph "packetLSTM vs Single LSTM" on page 8.
> >
> > The packetLSTM's novelty lies in enabling a dynamic ensemble of LSTMs to scale up and down as needed using a "novel, clean, and intuitive design," as appreciated by reviewer 1, which imparts packetLSTM the ability to handle haphazard inputs. Our design facilitates an effective balance between the local feature information and globally shared information, as corroborated by reviewer 1, by splitting these roles between individual LSTMs and a common long-term memory, which may not be possible in a direct adaptation of LSTM.

---

> > ### Author Response · Authors · 2024-11-16
> > **Reply to Weakness 3**
> >
> > **W.3.**  In Table 2, the authors indicate that the Max operator yields the best performance as the aggregation (AGG) function. However, this approach may overlook valuable information from other dimensions. Including an ablation study with a self-attention mechanism in the AGG operator, which may potentially enhance the aggregation of information across different features.
> >
> > **Answer:** We thank the reviewer for this suggestion and agree with the motivation to use self-attention for aggregation and that it may allow the model to also consider information from other dimensions.
> >
> > We have considered this possibility in the past but decided not to pursue it yet. The main reason is that our current knowledge of the self-attention mechanism indicates that the number of aggregate dimensions in the self-attention is fixed, which is inherently contradictory to the sixth characteristic of haphazard inputs, as explained below.
> >
> > Mathematically, self-attention learns three different weight matrices ($W^Q, W^K, W^V$) to determine query ($Q$), key ($K$), and value ($V$) vectors as $Q = XW^Q$, $K = XW^K$, and $V = XW^V$. Consequently, attention scores (A) is calculated as $A = \frac{QK^T}{\sqrt{d_k}}$, where $d_k$ is the dimension of the key vectors. The attention scores are further normalized using a softmax function as $\alpha = \frac{\exp(A)}{\sum \exp(A)}$. Finally, the aggregation embedding can be given by $Z = \alpha V$. Note that $X = [X_1, X_2, ..., X_n]$ here represents the dimensions to aggregate, and the number of dimensions is $n$, which is a fixed value. However, in the case of haphazard inputs, the value of $n$ can change at each instance. Therefore, self-attention cannot be employed directly as an aggregation operator in the haphazard inputs. If the maximum number of features is known, then self-attention may be adapted by setting the $X_i$ and $\alpha_i$ to 0 whenever the corresponding feature is not observed. However, in haphazard inputs, the maximum number of features is unknown; therefore, self-attention, to the best of our knowledge, cannot be employed as an aggregation operator.
> >
> > We thank the reviewer for pointing out the self-attention aggregation mechanism since we, too, believe it opens up a parallel line of research in the field of haphazard inputs and potentially in other fields that necessitate the variable number of aggregated dimensions, and the total number of dimensions is unknown.

---

> > ### Author Response · Authors · 2024-11-16
> > **Reply to Weakness 4**
> >
> > **W.4.** The paper would benefit from a more thorough theoretical analysis.
> >
> > **Answer:** We thank the reviewer for this suggestion and understand where the reviewer is coming from. We suppose the reviewer is seeking regret analysis conventionally done in online learning. However, the complexity of the deep learning method in the field of haphazard input creates a barrier to performing regret analysis, as we explain in the next paragraph.
> >
> > Performing regret analysis on the neural network (multi-layer perceptron) in an online learning setting entails several assumptions (reported in [1, 2]). Moreover, the regret analysis of LSTM in an online learning setting has not even been explored to the best of our knowledge. Most importantly, haphazard inputs introduce additional complexity of scalable architecture in the packetLSTM framework, where the number of parameters in the network changes in future instances. Therefore, it is not feasible to determine the regret bound of packetLSTM based on the current literature on regret analysis. This can be done in the future when the regret analysis of deep learning models is studied and established in detail in the field of haphazard inputs.
> >
> > Nonetheless, theoretical rigor is dear to us as well. Therefore, we have formulated the problem of haphazard inputs mathematically in section 3 on page 3. We have provided all our equations in complete detail to avoid potential theoretical gaps in Section 4, Figure 2, and Section D in the Appendix. We have also included a theoretical analysis of time and space complexity. If any additional theoretical analyses are required, we would appreciate it if the reviewer could point more specifically to the type of theoretical analysis they seek.
> >
> > [1] Gao, Ruiqi, Tianle Cai, Haochuan Li, Cho-Jui Hsieh, Liwei Wang, and Jason D. Lee. "Convergence of adversarial training in overparametrized neural networks." Advances in Neural Information Processing Systems 32 (2019).
> >
> > [2] Chen, Xinyi, Edgar Minasyan, Jason D. Lee, and Elad Hazan. "Regret guarantees for online deep control." In Learning for Dynamics and Control Conference, pp. 1032-1045. PMLR, 2023.

---

> > ### Author Response · Authors · 2024-11-16
> > **Reply to Weakness 5**
> >
> > **w.5.** The yellow color text in Figure 2 is too light, making it difficult to read clearly.
> >
> > **Answer:** We thank the reviewer for this comment. We have now darkened the yellow-colored text in Figure 2 for better readability.

---

### Official Review · Reviewer_kbnk · 2024-11-03

**Soundness:** 2
**Presentation:** 3
**Contribution:** 1
**Rating:** 3
**Confidence:** 4

**Summary:**

The paper presents PacketLSTM, a dynamic LSTM-based framework for streaming data with changing feature dimensions. Each input feature has a dedicated LSTM unit, allowing PacketLSTM to handle missing or new features continuously. The architecture combines shared memory for global representation and short-term memory for local representation. PacketLSTM achieves reasonable results across datasets.

**Strengths:**

- The paper addresses a pressing issue in streaming data analysis—handling temporal hazard inputs or dynamic features—which is crucial for the research community.
- The authors include a comprehensive set of experiments, including ablation studies and scenarios involving sudden and obsolete features, to verify the robustness of their proposed model.

**Weaknesses:**

- The paper does not implement several existing techniques for handling dynamic inputs within LSTM models. To establish PacketLSTM as the best choice, additional comparisons with implemented baseline methods for hazard input processing are necessary.
-  The proposed method, while interesting, may lack sufficient complexity and novelty to meet ICLR's technical bar.
- Performance improvements in the tables show small margin gains over other methods, so significance testing should be conducted to verify the robustness of these results.

**Questions:**

See weakness.
Some additional questions:
- Have the authors considered additional baseline implementations for handling dynamic inputs within LSTM models to strengthen their claim of superiority?
- How does PacketLSTM handle the computational complexity introduced by creating separate LSTMs for each feature? Is there a computational cost analysis for real-time applications?

---

> ### Author Response · Authors · 2024-11-16
> **Thanking reviewer for their feedback**
>
> We thank the reviewer for their feedback. We appreciate that the reviewer accepts the need for research in the field of haphazard inputs. We are glad that the reviewer recognized our extensive evaluation to demonstrate the robustness of the proposed method.
>
> We provide answers to all the reviewer's comments below and reflect them in our updated manuscript in blue color wherever applicable. We hope we were able to provide satisfactory answers. Please feel free to ask if the reviewer requires any further clarification.

---

> > ### Author Response · Authors · 2024-11-16
> > **Reply to Weakness 1 and Question 1**
> >
> > **W.1. and Q.1.** The paper does not implement several existing techniques for handling dynamic inputs within LSTM models. To establish PacketLSTM as the best choice, additional comparisons with implemented baseline methods for hazard input processing are necessary.
> >
> > **Answer:** We greatly value your advice on this aspect. Haphazard inputs are varying feature spaces, where the number of input features can change at each time step, and the total number of features is unknown. LSTM requires a fixed feature space, which necessitates the availability of all features. This requirement can be met through imputation, although it results in contradicting the sixth characteristic of haphazard inputs where the total number of features is unknown. We have discussed this in the paragraph "packetLSTM vs Single LSTM" on page 8.
> >
> > Despite our best efforts, we were unable to find any LSTM-based models that can handle haphazard inputs. This was one of our core motivations for proposing packetLSTM (lines 109-113).  We would greatly appreciate it if the reviewer could suggest 1 or 2 papers/techniques that are LSTM-based and can handle haphazard inputs. If feasible within the rebuttal timeline, we will also happily include them as a baseline in our manuscript.

---

> > ### Author Response · Authors · 2024-11-16
> > **Reply to Weakness 2**
> >
> > **W.2.** The proposed method, while interesting, may lack sufficient complexity and novelty to meet ICLR's technical bar.
> >
> > **Answer:** We gently beg to differ. A complex architecture is likely not an ideal solution for this problem as complex architecture presents rigidity, which may be a barrier to model haphazard inputs. The simplicity of our architecture imparts agility and an elegant mechanism to scale up and down as needed. Our novel "clean and intuitive design" was also noted by reviewer 1. Also, our contribution to the "dynamic handling of varying input dimensions through activation/deactivation of LSTMs" is appreciated by reviewers 1 and 2. Further, the "effective balance between local feature information and globally shared information" in our architecture imparts the ability to learn without forgetting.

---

> > ### Author Response · Authors · 2024-11-16
> > **Reply to Weakness 3**
> >
> > **W.3.** Performance improvements in the tables show small margin gains over other methods, so significance testing should be conducted to verify the robustness of these results.
> >
> > **Answer:** We appreciate your point about the robustness of results, for which we provided standard deviations of all our results in each table except for Table 2. For the results in Table 2, the standard deviation was provided in Table 9 in the Appendix due to space constraints and better readability. Additionally, the \% improvement of packetLSTM over the previously best-performing baseline ($pbb$) was provided in the last column of Table 1, which demonstrates that packetLSTM achieves at least 9.44\%, 3.50\%, 0.41\%, and 2.05\% improvement over $pbb$ on magic04, imdb, susy, and higgs datasets, respectively.
> >
> > We now also perform paired t-tests at 99\% confidence interval to determine the statistical significance of packetLSTM performance over $pbb$. We observed that the packetLSTM has a statistically significant better performance than all the models across each dataset except in a8a with p = 0.25 and 0.75. We have now included the paired t-tests in the paragraph "Results" on Page 6.
> >
> > We hope that the paired t-test results complement the percentage improvement numbers in Table 1, thus verifying the robustness of our results.

---

> > ### Author Response · Authors · 2024-11-16
> > **Reply to Question 2**
> >
> > **Q.2.** How does PacketLSTM handle the computational complexity introduced by creating separate LSTMs for each feature? Is there a computational cost analysis for real-time applications?
> >
> > **Answer:** We provide a theoretical analysis of the time and space complexity of packetLSTM in the paragraph "Complexity Analysis" on page 8. We further support the theoretical analysis with empirical evidence on page 26 in the section "Complexity Analysis." We provide the time required on each dataset by all the models (packetLSTM and baselines) in Table 8 on page 23.
> >
> > In the section "Complexity Analysis" on page 26, line 1392, we also provide the number of learnable parameters in each dataset corresponding to the worst-case scenario where all the features are present -- $\sim$183K, $\sim$131M, $\sim$2M, $\sim$148K, and $\sim$375K learnable parameters for magic04, imdb, a8a, SUSY, and HIGGS, respectively. Consequently, packetLSTM requires 1.40 MB, 999.45 MB, 15.26 MB, 1.13 MB, and 2.85 MB memory with 64-bit precision for magic04, imdb, a8a, SUSY, and HIGGS, respectively.
> >
> > Considering that a small large language model of 1 billion parameters is common nowadays, packetLSTM can handle $\sim$57K features with 1 billion parameters. We believe that a practical application would have a number of features way less than $\sim$57K. We have now included the above discussion in the paragraph "Number of Parameters" on page 26.

---

### Official Review · Reviewer_1RrH · 2024-11-03

**Soundness:** 3
**Presentation:** 4
**Contribution:** 2
**Rating:** 6
**Confidence:** 4

**Summary:**

The authors propose a dynamic LSTM-based online learning model called packetLSTM, designed to model streaming data with varying dimensionality. The core idea of the packetLSTM model is that each data feature corresponds to a separate LSTM network, which addresses the issue of changing feature dimensions. The authors conduct comparative experiments on multiple datasets, evaluating the performance of the packetLSTM model against several baseline models.

**Strengths:**

The authors propose a dynamic LSTM-based online learning model called packetLSTM, designed to model streaming data with varying dimensionality. The core idea of the packetLSTM model is that each data feature corresponds to a separate LSTM network, which addresses the issue of changing feature dimensions. The authors conduct comparative experiments on multiple datasets, evaluating the performance of the packetLSTM model against several baseline models.

**Weaknesses:**

1. The core design of packetLSTM, where each feature corresponds to an individual LSTM network, raises concerns about the model's practical applicability and scalability. While the authors provide a theoretical analysis of the spatial and temporal complexity of packetLSTM, there is a lack of specific experimental data to support these claims, such as memory consumption and training time. This absence of empirical evidence may lead readers to question the true potential of the model in real-world applications.

**Questions:**

1. The authors mention that "packetLSTM easily handles the 7500 features in the IMDb dataset." However, it is unclear what these 7500 features refer to, as most samples in the IMDb dataset do not have that many features. Do the 7500 features represent the total word count across all samples, or has the dataset undergone specific processing to yield this number? Could the authors clarify this point?

2. The authors claim that packetLSTM effectively prevents catastrophic forgetting. However, it appears that the experimental design does not sufficiently validate this assertion. Specifically, when features that have not been seen for a long time reappear, does the model genuinely maintain comparable performance? Could the authors provide additional evidence or analysis to support this claim?

3The authors reference the results from the OCDS paper, which report accuracies exceeding 80% on the IMDb, A8A, and Magic04 datasets. However, the accuracy achieved by the authors in replicating the OCDS results on these datasets is significantly lower. What might explain this discrepancy in performance? Could the authors provide insights into the differences in experimental setup or any other factors that could contribute to this variation in accuracy?

4.he authors set the parameter p to 0.25, 0.50, and 0.75 to simulate different degrees of feature missingness. However, would it be beneficial for the authors to provide a baseline evaluation of packetLSTM with complete features (i.e., p=0)? This would allow for an assessment of packetLSTM's optimal performance when all features are available, serving as a reference point for interpreting the results under varying levels of feature absence.

5, I have concerns about the reliability of some experimental results presented in the paper. The authors mention that the balanced accuracy on the IMDb dataset for the DistilBERT, BERT, and OCDS models is only slightly above 50%. Given that IMDb is a binary classification dataset, this low balanced accuracy may indicate that the models are not effectively learning useful information, with performance levels approaching random guessing. What might explain these results? A balanced accuracy of 50% suggests that the mean sensitivity and specificity are both around 0.5, raising the possibility that the model may be making complete errors in one class while performing well in the other, which typically indicates significant bias. Additionally, since some datasets are inherently balanced, random feature missingness should theoretically have minimal impact on the dataset's balance. Would the authors consider providing additional metrics, such as standard accuracy, to further clarify model performance?

**Details Of Ethics Concerns:**

I wrote the comments in the wrong section

---

> ### Author Response · Authors · 2024-11-16
> **Thanking reviewer for their feedback**
>
> We thank the reviewer for their positive and constructive feedback. We are glad to know that the reviewer liked our paper.
>
> We provide answers to all the reviewer's comments below and reflect them in our updated manuscript in blue color wherever applicable. We hope we were able to provide satisfactory answers. Please feel free to ask if the reviewer requires any further clarification.

---

> > ### Author Response · Authors · 2024-11-16
> > **Reply to Weakness 1**
> >
> > **W.1.** The core design of packetLSTM, where each feature corresponds to an individual LSTM network, raises concerns about the model's practical applicability and scalability. While the authors provide a theoretical analysis of the spatial and temporal complexity of packetLSTM, there is a lack of specific experimental data to support these claims, such as memory consumption and training time. This absence of empirical evidence may lead readers to question the true potential of the model in real-world applications.
> >
> > **Answer:** We understand the reviewer's concern and agree that empirical evidence is helpful. While we did include empirical evidence in terms of actual time taken (Table 8), we could have also included the storage memory required by the packetLSTM for each dataset.
> >
> > Additionally, in order to indicate the potential of packetLSTM in real-world applications, we now show that packetLSTM can handle $\sim$57K features with 1 billion parameters, which represents the size of a small large language model (LLM). Considering that $\sim$57K features are sufficient for most large-scale practical applications and packetLSTM can handle that with 1 billion parameters, we believe that packetLSTM will not face challenges in diverse real-world applications.
> >
> > **Empirical evidence of time complexity:** The space and time complexity of packetLSTM are mentioned in the paragraph "Complexity Analysis" on page 8 and further expanded in section P titled "Complexity Analysis" on page 26, where we empirically outline the time taken by packetLSTM. We also provide the time requirement of packetLSTM and baselines in Table 8. We would like to point out that packetLSTM is the fastest among the deep learning methods (Aux-Net and Aux-Drop). We further demonstrate that the difference between the time required by packetLSTM for different $p$ values is not huge, demonstrating scalability in terms of time complexity corresponding to the number of features.
> >
> > **Empirical evidence of space complexity:** We also provide the number of learnable parameters in packetLSTM for each dataset in the paragraph "Number of Parameters" on page 26, which is $\sim$183K, $\sim$131M, $\sim$2M, $\sim$148K, and $\sim$375K learnable parameters for magic04, imdb, a8a, SUSY, and HIGGS, respectively. Consequently, packetLSTM requires 1.40 MB, 999.45 MB, 15.26 MB, 1.13 MB, and 2.85 MB memory with 64-bit precision for magic04, imdb, a8a, SUSY, and HIGGS, respectively. Considering that a small large language model of 1 billion parameters is common nowadays, packetLSTM can handle $\sim$57K features with 1 billion parameters. We believe that a practical application would have a number of features way less than $\sim$57K. We have now included the above discussion in the paragraph "Number of Parameters" on page 26.

---

> > ### Author Response · Authors · 2024-11-16
> > **Reply to Question 1**
> >
> > **Q.1** The authors mention that "packetLSTM easily handles the 7500 features in the IMDb dataset." However, it is unclear what these 7500 features refer to, as most samples in the IMDb dataset do not have that many features. Do the 7500 features represent the total word count across all samples, or has the dataset undergone specific processing to yield this number? Could the authors clarify this point?
> >
> > **Answer:** We follow the previous literature in the field of haphazard inputs [1, 2] to prepare the imdb dataset. This approach adheres to the standard practices of previous works and is explained briefly here. The original imdb dataset consists of training and test subsets, each containing 25000 instances. Following the previous literature [1,2], we use only the training subset. Within the training subset, 7500 most prevalent words are used as the 7500 features following the prescription in [1,2]. We now include the above information in Section E on page 19.
> >
> > [1] Ege Beyazit, Jeevithan Alagurajah, and Xindong Wu. Online learning from data streams with varying feature spaces. In AAAI Conference on Artificial Intelligence, volume 33, pp. 3232–3239,
> > 2019.
> >
> > [2] Rohit Agarwal, Arijit Das, Alexander Horsch, Krishna Agarwal, and Dilip K Prasad. Online learning under haphazard input conditions: A comprehensive review and analysis. arXiv preprint arXiv:2404.04903, 2024.

---

> > ### Author Response · Authors · 2024-11-16
> > **Reply to Question 2**
> >
> > **Q.2** The authors claim that packetLSTM effectively prevents catastrophic forgetting. However, it appears that the experimental design does not sufficiently validate this assertion ...
> >
> > **Answer:** We thank the reviewer for this comment. We realize that even though we provided an experiment to support this claim in Figure 4, we could have included 'catastrophic forgetting' in the figure caption for better assimilation, and it would have helped to direct the reader to the appendix for more details.
> >
> > We understood the importance of a detailed experiment on this aspect. Therefore, we included explicit details of our experiment, numerical values, and discussion in Section S and Tables 11-14 in the Appendix. We have now expanded section S.1 even more to include explicit observations to guide the reader towards the outcome of the experiment.
> >
> > Furthermore, we present the experiment in detail here for the reviewer's immediate perusal.
> >
> > ## Experiment setup
> >
> > The experiment in paragraph "Reappearing" on page 9 is designed to observe if a model forgets the learned information from features when those features are not observed for a long time.
> >
> > Therefore, we created two groups of features -- the first group containing the first 50\% of total features and the second group comprising the last 50\% of the total features. We considered the HIGGS and SUSY datasets for this experiment due to their substantial number of instances (1 million instances). The feature groups do not appear for a long time gap (0.2 million instances), allowing the model to catastrophically forget its learning. Specifically, the 1 million instances are divided into five intervals of 0.2 million instances each -- the first interval contains the first 0.2 million instances, the second interval consists of the next 0.2 million instances, and so on. The first group of features arrives in the first, third, and fifth intervals. Therefore, the first group of features is absent at the second and fifth intervals.
> >
> > If a model has a risk of catastrophic forgetting, then the information learned from the first group of features in the first interval will be forgotten during the second interval. This would result in lower performance of the model in the third interval when the first group of features is observed again. However, if the model can learn without forgetting, it will maintain or improve its performance in the third interval. The same idea is applicable in the fifth interval for the first group of features and the fourth interval for the second group of features.
> >
> > We utilize the mean balance accuracy of five runs in each interval to determine the performance increase or decrease, and these values, along with the standard deviation, are provided in Tables 11-14.
> >
> > ## Observation
> >
> > The observation on the HIGGS dataset is presented in Figure 4 and the paragraph titled "Reappearing" on page 9 in the main manuscript. Similar observations are also drawn in the SUSY dataset and are provided in Figure 7 and the paragraph titled "Reappearing" on page 30 in the Appendix. The best baseline method in the HIGGS and SUSY dataset is Aux-Drop (see Table 1). Therefore, we present the observation from the Aux-Drop point of view, and the same applies to other baselines.
> >
> > 1. **Aux-Drop:** The performance of Aux-Drop decreased from 53.52 in the first interval to 52.98 in the third interval. The performance further decreased to 50.93 in the fifth interval. Aux-Drop also showed a performance decrease from 55.48 in the second interval to 52.38 in the fourth interval. Therefore, Aux-Drop suffers from catastrophic forgetting.
> >
> > 2. **packetLSTM:** The performance of packetLSTM increases from 58.74 in the first interval to 60.42 in the third interval. The performance further increases to 60.54 in the fifth interval. The packetLSTM also showed a performance increase from 53.27 in the second interval to 53.94 in the fourth interval. Therefore, we argue that packetLSTM mitigates catastrophic forgetting.
> >
> > ## Further evidence of packetLSTM mitigating catastrophic forgetting
> >
> > **Counterargument to packetLSTM preventing catastrophic forgetting:** It can be argued that the performance increase of packetLSTM in the third interval is not due to mitigation of catastrophic forgetting; it is instead due to a likely situation of better predictive capability of the features present in the third interval compared to the first interval.
> >
> > **Refutation:** To refute the above argument, we retrained the packetLSTM in each interval -- a new initialized packetLSTM was considered for each interval -- and referred to it as packetLSTM-Retraining (see Figure 4 and Table 12). The performance of the packetLSTM (60.42) outperforms packetLSTM-Retraining (58.81) in the third interval, which shows that packetLSTM can retain its learning from the first interval. Similar performance increase is observed in the fourth and fifth intervals.

---

> > ### Author Response · Authors · 2024-11-16
> > **Reply to Question 3**
> >
> > **Q.3.** The authors reference the results from the OCDS paper, which report accuracies exceeding 80\% on the IMDb, A8A, and Magic04 datasets. However, the accuracy achieved by the authors in replicating the OCDS results on these datasets is significantly lower. What might explain this discrepancy in performance? Could the authors provide insights into the differences in experimental setup or any other factors that could contribute to this variation in accuracy?
> >
> > **Answer:** We would like to point out that the official code of the OCDS [1] is not publicly available. The only article that provides the code of the OCDS paper is [2]. Therefore, we followed the implementation provided in [2] for OCDS, which is also stated in line 1130. One of the problems observed in the OCDS and mentioned in [2] on page 8 is:
> >
> > > *The OCDS algorithm updates weights according to the equation $w_{t+1} = w_t - \nabla_{w_t}\mathcal{F}$, where $\nabla_{w_t}\mathcal{F} = -2(y_t - w_t^\top \psi(x_t))\psi(x_t) + \beta_1 \partial||w_t||_1 +  \beta_2 (L + L^\top)w_t$.
> > Here $\beta_1$ and $\beta_2$ are scale parameters to absorb the magnitudes of $\partial||w_t||_1$ and $(L + L^\top)w_t$, respectively. Similarly, we observe that the first term ($-2(y_t - w_t^\top \psi(x_t))\psi(x_t)$), frequently exceeds acceptable bounds for many datasets. To counteract this, we introduce an absorption scale hyperparameter ($\beta_0$) to moderate the value of the first term*
> >
> > Therefore, it might be possible that the datasets went through some kind of pre-processing in the OCDS paper. However, we cannot confirm this since there is no mention of any pre-processing in the paper, and the paper does not provide any official code.
> >
> > [1] Yi He, Baijun Wu, Di Wu, Ege Beyazit, Sheng Chen, and Xindong Wu. Online learning from capricious data streams: a generative approach. In International Joint Conference on Artificial Intelligence, Main track, 2019.
> >
> > [2] Rohit Agarwal, Arijit Das, Alexander Horsch, Krishna Agarwal, and Dilip K Prasad. Online learning under haphazard input conditions: A comprehensive review and analysis. arXiv preprint arXiv:2404.04903, 2024.

---

> > ### Author Response · Authors · 2024-11-16
> > **Reply to Question 4**
> >
> > **Q.4.** The authors set the parameter p to 0.25, 0.50, and 0.75 to simulate different degrees of feature missingness. However, would it be beneficial for the authors to provide a baseline evaluation of packetLSTM with complete features (i.e., p=0)? This would allow for an assessment of packetLSTM's optimal performance when all features are available, serving as a reference point for interpreting the results under varying levels of feature absence.
> >
> > **Answer:** We thank the reviewer for this suggestion. Probably, the reviewer meant $p = 1$ to indicate the complete features. As per the reviewer's suggestion, we set $p = 1$ in the synthetic datasets to determine the upper bound performance of packetLSTM when all the features are present at each time instance. The corresponding results are provided in Table 16 on page 33 and in this comment for easy reference. We compare packetLSTM's performance across varying $p$ values (0.25, 0.5, 0.75, and 1) in Figure 8 on page 33. Unsurprisingly, the performance of packetLSTM increases with decreasing haphazardness (i.e., increasing $p$ value) in the data. The most significant decline in performance occurs at $p= 0.25$, which is expected due to the increased haphazardness. We now include this experiment in section W in the Appendix on page 33.
> >
> > | Metric | magic04          | a8a              | SUSY                | HIGGS              |
> > |--------|------------------|------------------|---------------------|--------------------|
> > | bAcc   | 79.65$\pm$0.13   | 73.57$\pm$0.15   | 77.42$\pm$0.05      | 62.66$\pm$0.18     |
> > | Acc    | 83.68$\pm$0.12   | 83.37$\pm$0.08   | 78.11$\pm$0.05      | 62.98$\pm$0.19     |
> > | ROC    | 87.79$\pm$0.06   | 87.54$\pm$0.11   | 84.92$\pm$0.05      | 67.95$\pm$0.28     |
> > | PRC    | 90.49$\pm$0.11   | 95.49$\pm$0.06   | 85.17$\pm$0.06      | 69.60$\pm$0.24     |
> > | Err    | 3103.2$\pm$21.93 | 5415.4$\pm$26.19 | 218943.6$\pm$492.38 | 370198$\pm$1863.46 |
> > | Time   | 79.02$\pm$8.97   | 261.74$\pm$14.66 | 4146.19$\pm$98.9    | 5589.51$\pm$94.65  |

---

> > ### Author Response · Authors · 2024-11-16
> > **Reply to Question 5**
> >
> > **Q.5.** I have concerns about the reliability of some experimental results presented in the paper. The authors mention that the balanced accuracy on the IMDb dataset for the DistilBERT, BERT, and OCDS models is only slightly above 50\%. Given that IMDb is a binary classification dataset, this low balanced accuracy may indicate that the models are not effectively learning useful information, with performance levels approaching random guessing. What might explain these results? A balanced accuracy of 50\% suggests that the mean sensitivity and specificity are both around 0.5, raising the possibility that the model may be making complete errors in one class while performing well in the other, which typically indicates significant bias. Additionally, since some datasets are inherently balanced, random feature missingness should theoretically have minimal impact on the dataset's balance. Would the authors consider providing additional metrics, such as standard accuracy, to further clarify model performance?
> >
> > **Answer:** We thank the reviewer for their comment and observation. We agree with the reviewer's suggestion that additional metrics are also needed to further clarify model performance.
> >
> > We report 5 metrics, namely, balanced accuracy, standard accuracy, AUROC, AUPRC, and the number of errors for benchmarking in our paper. These metrics are reported in Table 8 in the Appendix because of the page limit. We chose balanced accuracy for comparison in Table 1 in the main manuscript because most datasets (4 out of 5) are imbalanced. As per the reviewer's suggestion, we now also provide the values of the 5 metrics for DistilBERT and BERT in Table 15 in the Appendix.
> >
> > We also agree with the reviewer that in the imdb dataset (a balanced dataset), DistilBERT, BERT, and OCDS models are possibly random guessing since the AUROC and standard accuracy is also approximately 50\% (see Table 8 for OCDS and Table 15 for DistilBERT and BERT).

---

### Official Review · Reviewer_k3D3 · 2024-11-04

**Soundness:** 4
**Presentation:** 4
**Contribution:** 3
**Rating:** 8
**Confidence:** 4

**Summary:**

The paper introduces packetLSTM, a dynamic LSTM-based framework designed to handle streaming data with varying input feature dimensions in online learning settings. The key innovation is using a packet of LSTMs, where each LSTM is dedicated to processing one input feature, along with a shared common memory for global information. The framework can dynamically activate, deactivate, and add new LSTMs as features appear or disappear. The authors demonstrate superior performance over existing methods across 5 datasets and extend their approach to other RNN architectures (GRU, vanilla RNN). They also introduce a new Transformer-based baseline called HapTransformer.

**Strengths:**

1. The authors propose novel architecture about their models with clean and intuitive design using one LSTM per feature with shared memory, dynamic handling of varying input dimensions through activation/deactivation of LSTMs, and effective balance between local feature information and global shared information
2. The evaluation in the experiments is comprehensive. For exmaple, the authors use 5 diverse datasets, detailed ablation studies examining each component, strong empirical results consistently outperforming 10 baselines. and well-designed challenging scenarios testing model capabilities.
3. The complexity analysis is also beneficial and brings more applicability of the model. The proposed HapTransformer is also a novel contribution in the hot transformer domain.
4. The extensive hyperparameter search protocols is also promising.

**Weaknesses:**

1. Some important literatures about varying input feature dimensions are missing. The authors are encouraged to include many fundamental works from this domain. For example, the authors should cite [1] and [2] and several other references from [3].
2. This work focuses primarily on binary classification tasks. It would be better to see more diverse task such as multi-class classification and regression problem. No need to bring forward such results during rebuttal but I would like to see some of them in the final version.

[1] Hou, Bo-Jian, Lijun Zhang, and Zhi-Hua Zhou. "Learning with feature evolvable streams." Advances in Neural Information Processing Systems 30 (2017).

[2] Hou, Chenping, and Zhi-Hua Zhou. "One-pass learning with incremental and decremental features." IEEE transactions on pattern analysis and machine intelligence 40, no. 11 (2017): 2776-2792.

[3] He, Y., Schreckenberger, C., Stuckenschmidt, H. and Wu, X., 2023, August. Towards Utilitarian Online Learning-A Review of Online Algorithms in Open Feature Space. In IJCAI (pp. 6647-6655).

**Questions:**

1. Given the space complexity challenges, what are the practical limits on the number of features the model can handle efficiently?
2. Could the approach be extended to handle structured data types (e.g., graphs, images) where features have inherent relationships?

---

> ### Author Response · Authors · 2024-11-16
> **Thanking reviewer for their feedback**
>
> We thank the reviewer for positive and constructive feedback. We are glad that the reviewer liked our paper. We highly appreciate the reviewer's recognition of the novelty of our proposed method and the extensive benchmarking and evaluation performed by us.
>
> We provide answers to all the reviewer's comments below and reflect them in our updated manuscript in blue color wherever applicable. We hope we were able to provide satisfactory answers. Please feel free to ask if the reviewer requires any further clarification.

---

> > ### Author Response · Authors · 2024-11-16
> > **Reply to Weakness 1**
> >
> > **W.1.** Some important literatures about varying input feature dimensions are missing. The authors are encouraged to include many fundamental works from this domain. For example, the authors should cite [1] and [2] and several other references from [3].
> >
> > **Answer:** We thank the reviewer for highlighting the additional references.
> >
> > We initially did not include references [1] and [2] because they assume some form of structure in their data stream (feature evolvable streams [1] and incremental and decremental features [2]), which contradicts the characteristics of haphazard inputs.
> >
> > Nonetheless, we agree with the reviewer that these works merit discussion. We now provide a discussion of [1] and [2] along with two more appropriate citations from [3] in Section V of the Appendix on page 33, where we outline the differences between haphazard inputs compared to feature evolvable streams and incremental and decremental features. This discussion is referenced in lines 101-102 in the main manuscript.

---

> > ### Author Response · Authors · 2024-11-16
> > **Reply to Weakness 2**
> >
> > **W.2.** This work focuses primarily on binary classification tasks. It would be better to see more diverse task such as multi-class classification and regression problem. No need to bring forward such results during rebuttal but I would like to see some of them in the final version.
> >
> > **Answer:** We thank the reviewer for this suggestion and appreciate their consideration in providing us time till the final manuscript deadline to include results on other tasks.
> >
> > We adhered to binary classification because it is the standard task in the field of haphazard inputs.  However, we agree that it would benefit to expand the field of haphazard inputs by including diverse tasks like multi-class classification and regression.
> >
> > We will consider a few datasets from the UCI repository and transform them into haphazard inputs following the process outlined in the paragraph "Synthetic Dataset Preparation" on page 19. We will benchmark packetLSTM along with the baseline models and include the results in the Appendix.

---

> > ### Author Response · Authors · 2024-11-16
> > **Reply to Question 1**
> >
> > **Q.1.** Given the space complexity challenges, what are the practical limits on the number of features the model can handle efficiently?
> >
> > **Answer:** The packetLSTM can handle $\sim$57K features with 1 billion parameters, as explained in the next paragraph. Considering that $\sim$57K features are sufficient for most large-scale practical applications and packetLSTM can handle that with 1 billion parameters, we believe that its space complexity will not be a barrier. Note that in the current scenario, the 1 billion parameter corresponds to the size of a small large language model.
> >
> > The number of learnable parameters in packetLSTM is **$\boldsymbol{|\mathbb{F}^t|*(4s^2 + 17s) + 2s^2 + 4s + 2}$**, where $|\mathbb{F}^t|$ is the number of features available at time $t$. For all the experiments in our work, the value of $s$ is 64. Therefore, the number of learnable parameters is $17472*|\mathbb{F}^t| + 8450$. The worst-case scenario corresponds to the situation where all the features are present. Therefore, packetLSTM can handle $\sim$57K features, which correspond to 1 billion parameters.
> >
> > We have now included the practical limits of packetLSTM in the paragraph "Number of Parameters" on page 26, lines 1395-1397.

---

> > ### Author Response · Authors · 2024-11-16
> > **Reply to Question 2**
> >
> > **Q.2.** Could the approach be extended to handle structured data types (e.g., graphs, images) where features have inherent relationships?
> >
> > **Answer:** This is an interesting question that we are also working on. While it may not be possible to apply packetLSTM to structured data (graphs and images) in the current form, we can adapt packetLSTM for structured data in two possible manners, as explained in the next paragraphs.
> >
> > Extending packetLSTM for graphs and images would mean considering individual LSTMs for each node and pixel. Although packetLSTM facilitates information sharing and collaborative learning across all the features through a global summary state, this may not be the best solution for graphs and images where nodes and pixels have inherent and direct relationships with other neighboring nodes and pixels. In the future, we may incorporate interconnection between LSTM blocks, allowing packetLSTM to preserve the inherent structural relationships.
> >
> > Alternatively, packetLSTM can be employed for multi-modal learning where the modalities may correspond to graphs and images, and appropriate networks can be used to transform the graphs and images into embeddings. These embeddings can then be used as inputs to the packetLSTM. Thereon, packetLSTM, through its scalable and dynamic architecture, can easily handle multi-modal data with missing modalities or a variable number of modalities.

---

### Meta-Review · Area_Chair_LJVS · 2024-12-24

**Metareview:**

**Summary:** This paper proposes packetLSTM, a novel framework to handle streaming data with varying input dimensions in online learning settings. The method employs a packet of LSTMs, each corresponding to one input feature, with a shared global memory for consolidating information. The model dynamically activates or deactivates LSTMs as features appear or disappear, aiming to mitigate catastrophic forgetting. Experimental results on five datasets demonstrate competitive or superior performance compared to baselines, with extensive ablation studies included. The authors also introduce HapTransformer as a new baseline.

**Decision:** This is a borderline paper. Reviewers noted the significance of the proposed architecture to varying input dimensions in streaming data, as well as solid empirical analysis. In addition, the authors also addressed several concerns, for example, about the scalability of the method, and review of related work. However, the critical concern about the novelty of packetLSTM persists, and the authors also failed to provide additional empirical evaluations on multi-class or regression tasks. In addition, reviewers also raised concerns about the lack of theoretical insights of this work. These concerns collectively lead to the decision to reject.

**Additional Comments On Reviewer Discussion:**

During the discussion phase, the authors addressed several concerns by including additional analyses and results. They provided updates on memory and computational requirements, and improved clarity in presenting catastrophic forgetting, as well as some key design elements. However, these updates were not sufficient to alleviate the fundamental weaknesses. Concerns about limited novelty and insufficient task diversity persisted among reviewers. Despite the authors’ efforts, the reviewers largely maintained their initial assessments, leading to the decision to reject. During the reviewer-AC discussion, no objections were raised to this decision.

---

### Decision · Program_Chairs · 2025-01-22

Reject